# THE NEEDLE IN THE HAYSTACK: OUT-DISTRIBUTION AWARE SELF-TRAINING IN AN OPEN-WORLD SETTING

## ABSTRACT

Traditional semi-supervised learning (SSL) has focused on the closed world assumption where all unlabeled samples are task-related. In practice, this assumption is often violated when leveraging data from very large image databases that contain mostly non-task-relevant samples. While standard self-training and other established methods fail in this open-world setting, we demonstrate that our out-distribution-aware self-learning (ODST) with a careful sample selection strategy can leverage unlabeled datasets with millions of samples, more than 1600 times larger than the labeled datasets, and which contain only about $2\%$ task-relevant inputs. Standard and open world SSL techniques degrade in performance when the ratio of task-relevant sample decreases and show a significant distribution shift which is problematic regarding AI safety while ODST outperforms them with respect to test performance, corruption robustness and out-of-distribution detection.

## 1 INTRODUCTION

In past years we have seen tremendous progress in image recognition based on deep learning (Krizhevsky et al., 2012; He et al., 2016). However, this success required large labeled datasets that are expensive to generate. On the other hand, large amounts of unlabeled data are broadly available, in particular in image recognition. The promise of semi-supervised learning (Chapelle et al., 2006) is to leverage unlabeled data in order to improve prediction performance. However, the underlying assumption of traditional and modern (Berthelot et al., 2019; Sohn et al., 2020) SSL algorithms is that the unlabeled data comes from the same distribution or at least contains data from the same classes (closed world assumption). Oliver et al. (2018) criticized this as being unrealistic as the assumption is hard to check when retrieving large amounts of unlabeled data from the web. SSL in an open world setting, where the unlabeled data contains task-relevant but also non-related out-of-distribution (OOD) images, has recently attracted attention as a more realistic approach to SSL (Guo et al., 2020; Chen et al., 2020; Yu et al., 2020). While these methods could outperform standard SSL techniques when the unlabeled data contains out-of-distribution samples, they have mostly been evaluated in settings where the unlabeled data contains relatively few non-task-related samples. It is thus an open question whether existing open world SSL methods can scale to large unlabeled data bases where the ratio of task-relevant to non-related samples is much smaller.

In this paper, we demonstrate that existing methods suffer from severe performance degradations when the ratio of non-related images in the unlabeled dataset increases. Moreover, even if they achieve high accuracy, they are not OOD aware, that is they systematically assign high confidence to non-related inputs, e.g. a CIFAR10 model classifies images containing humans as dogs or horses. Our contributions are: i) we propose our scalable, iterative out-distribution-aware self-training (ODST) which enforces low-confidence predictions on non-task-related samples and selects unlabeled samples for the in-distribution task via a novel class-adaptive selection scheme. This confidence based selection scheme allows us to deal with strongly unbalanced in-distribution classes in the unlabeled dataset, ii) we outperform state-of-the-art SSL techniques (Berthelot et al., 2019; Sohn et al., 2020), as well as recent open world SSL techniques (Guo et al., 2020; Yu et al., 2020) in terms of prediction performance and out-of-distribution detection and show that all other approaches suffer from distribution shifts, iii) we show strong performance gains when using ODST with the full training sets of CIFAR10 and CIFAR100 as labeled data together with the 80 Million Tiny Image (80MTI) dataset as unlabeled dataset. Thus we show that even with large amounts of labeled data, ODST can still leverage unlabeled data to improve prediction and OOD performance.

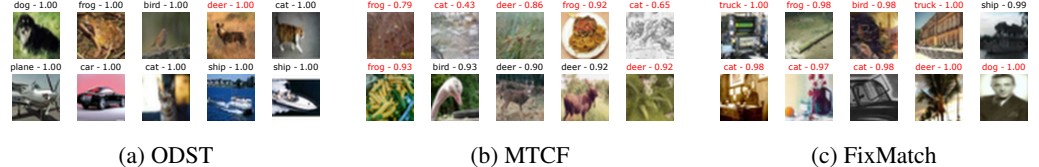

(a) ODST             (b) MTCF             (c) FixMatch

Figure 1: Random unlabeled samples considered to be task-relevant by the SSL method (confidence above in-distribution thresholds for ODST, above 95% confidence for Fixmatch and above the Otsu threshold for MTCF) are shown together with their confidence and predicted label (mistakes are marked red). MTCF and Fixmatch show severe distribution shift and only our ODST is able to select the correct samples. All methods are trained on CIFAR10 with 4k labeled images and an unlabeled set consisting of 41k CIFAR10 training images and 1M images from 80MTI.

## 2 RELATED WORK

**Semi-supervised learning (SSL)** is an established technique (Zhu, 2005; Zhu & Goldberg, 2009) for leveraging information from unlabeled data to improve predictive performance. In self-training (Riloff, 1996; Riloff & Wiebe, 2003; Scudder, 1965), a teacher model is trained in a fully-supervised fashion on a labeled dataset. The teacher model is then used to label a set of unlabeled examples, typically drawn from the original data distribution (closed world assumption), which is then used in combination with the labeled samples to train a new student model. Various extensions of this protocol have been proposed, including the use of an ensemble of teacher models (Zhou et al., 2018) and Co-training (Blum & Mitchell, 1998).

Recently, Xie et al. (2020) and Yalniz et al. (2019) used self-training to improve performance on ImageNet (Russakovsky et al., 2015) by using large image databases consisting of millions of task-relevant and out-of-distribution samples. On CIFAR10, Carmon et al. (2019) were able to significantly improve model robustness to adversarial perturbations by adding unlabeled samples from 80 million tiny images (80MTI) (Torralba et al., 2008) using self-training. The distinctive feature of self-training in comparison to other SSL methods is that the training of the teacher model is separated from the labeling process. In contrast, in pseudo-labeling (Lee, 2013; Iscen et al., 2019; Shi et al., 2018) labels are generated during training by the model itself. Similarly, consistency-based SSL-methods like Π-models (Laine & Aila, 2016; Sajjadi et al., 2016), mean-teacher (Tarvainen & Valpola, 2017) and virtual adversarial training (Miyato et al., 2018) enforce an invariance of the model's output on the unlabeled data under a specific set of perturbations. Methods like MixMatch (Berthelot et al., 2019) and FixMatch (Sohn et al., 2020) combine consistency regularization with strong augmentation e.g. RandAugment (Cubuk et al., 2020). A related technique is entropy-minimization (Grandvalet & Bengio, 2005), which *penalizes* low-confidence predictions on unlabeled samples during training. Oliver et al. (2018) found that SSL can improve the model's performance in the traditional SSL setting where the unlabeled data is sampled from the same distribution as the training data (closed world assumption) but can degrade the performance when the unlabeled data contains non-task-related samples (open world setting).

The open world SSL setting has has been explored only recently, by combining elements of consistency regularization with online OOD detection and sample filtering (Yu et al., 2020; Chen et al., 2020) or soft per-sample weighting (Guo et al., 2020). While they demonstrate performance improvements when the unlabeled data contains non-task-related samples, their evaluation is restricted to settings where the unlabeled data contains mostly task-related samples. We later demonstrate that in more challenging settings, where the ratio of task-related samples in the unlabeled data is small, these methods show severe performance degradations.

**Out-of-distribution detection:** Deep Neural networks (DNN) have empirically and theoretically been shown to produce overconfident predictions for inputs not related to the task, e.g. noise or other classes not contained in the labeled dataset (Nguyen et al., 2015; Hendrycks & Gimpel, 2017; Hein et al., 2019), i.e. the confidence of a DNN is not reliable for the detection of out-of-distribution (OOD) samples. Approaches for OOD detection include ODIN (Liang et al., 2018) or using the Mahalanobis distance of higher-order features (Lee et al., 2018). Hendrycks et al. (2019) introduced Outlier exposure (OE), see Hein et al. (2019) for the related CEDA, and show that the confidence

can be used as a reliable OOD detector when enforcing low confidence on an OOD training set, even when tested on other OOD test datasets. OOD detection is also related to open set recognition (Boult et al., 2019) which is beyond the scope of this paper.

## 3 METHOD

In contrast to previous works which focused on problems where the ratio of non-task-related to task-related images is small, our goal is to show that our ODST in combination with a careful sample selection strategy yields a self-training scheme that can leverage large unlabeled datasets to improve performance on the CIFAR10 and CIFAR100 test set over a fully-supervised baseline trained on the entire training set and additionally has excellent OOD detection performance.

First, we introduce self-training as in the noisy student self-training (NSST) of Xie et al. (2020) which serves a baseline. Then we introduce our out-distribution aware self-training ODST.

In the following $\mathbf{T} = (x_i, y_i)_{i=1}^n$ denotes the set of labeled examples, where $x_i \in \mathbb{R}^d$ and $y_i \in \{1, \ldots, K\}$ and $\mathbf{U} = (z_i)_{i=1}^m$ is a collection of unlabeled samples. The traditional SSL literature makes the assumption that the unlabeled samples $\mathbf{U}$ are drawn from the same distribution as the labeled examples $\mathbf{T}$, or at least belong to the same set of classes which we explicitly do not do in this paper. Given the logits of a neural network $f : \mathbb{R}^d \to \mathbb{R}$ the predicted probability distribution for a point $x$ is computed via the softmax as: $\hat{p}_f(s|x) = e^{f_s(x)} / \sum_{l=1}^K e^{f_l(x)}$. The confidence in the decision for $x$ is then given by $\max_{s=1,\ldots,K} \hat{p}_f(s|x)$, and the cross-entropy loss between (soft)-labels $p \in \mathbb{R}^K$ ($\sum_i p_i = 1$, $p_i \geq 0$) and prediction $\hat{p}$ is defined as: $L(p, \hat{p}) = -\sum_{i=1}^K p_i \log \hat{p}_i$.

### 3.1 NOISY STUDENT SELF-TRAINING (NSST)

In NSST (Xie et al., 2020) the first teacher is a base model $f^{(0)}$ obtained by minimizing the cross-entropy loss on the labeled set only. The iterative scheme of NSST starting at $t = 0$ is:

  1) pseudo-label all unlabeled samples in $\mathbf{U}$ with current teacher $f^{(t)}$

  2) select a subset $\mathbf{I} \subset \mathbf{U}$ of the pseudo-labeled examples by choosing per class the top-$k$ with highest confidence and which are above a fixed threshold

  3) train new model $f^{(t+1)}$ by minimizing the loss on labeled and pseudo-labeled samples in $\mathbf{I}$ with AutoAugment (Cubuk et al., 2019) as data augmentation:

$$\frac{1}{n} \sum_{i=1}^n L\big(y_i, \hat{p}_{f^{(t+1)}}(x_i)\big) + \frac{1}{|\mathbf{I}|} \sum_{z \in \mathbf{I}} L\big(\hat{p}_{f^{(t)}}(z), \hat{p}_{f^{(t+1)}}(z)\big)$$

  4) $t \leftarrow t + 1$ and go back to step 1

Xie et al. (2020) call their self-training noisy due to the very strong data augmentation which avoids overfitting via noise injection. The main difficulty in self-training in a closed world setting is the propagation of labeling mistakes which leads to a degradation of prediction performance which is taken care of by step 2). While Xie et al. (2020) chose a fixed threshold, we choose it according to the false positive rate on an in-distribution validation set. However, in an open-world setting, an equally severe problem is that a large fraction of the unlabeled samples is not task-relevant such that including them leads to a distribution shift which happens for NSST (see Figure 2) and other SSL methods (see Figure 1). However, more importantly, the classifier is highly confident on unrelated classes which is a problem for AI safety and this undesired behavior is even hard to notice as predictive performance might appear to improve when evaluated only on the test set.

### 3.2 OUT-DISTRIBUTION AWARE SELF-TRAINING (ODST)

We first provide an overview over our algorithmic scheme and then explain the individual steps in more detail. The base ODST model is initialized with an out-distribution aware base teacher model $f^{(0)}$, trained by minimizing:

$$\frac{1}{n} \sum_{i=1}^n L\big(y_i, \hat{p}_{f^{(0)}}(x_i)\big) + \frac{1}{|\mathbf{U}|} \sum_{z \in \mathbf{U}} L\Big(\frac{1}{K}\mathbf{1}, \hat{p}_{f^{(0)}}(z)\Big). \tag{1}$$

We then iterate the following steps starting from $t = 0$:

A) calibrate $f^{(t)}$ on the in-distribution validation set

B) pseudo-label all unlabeled samples in $\mathbf{U}$ with current teacher $f^{(t)}$

C) for each class $c$: select the top-$k$ unlabeled instances with highest confidence classified as $c$ that lie above the threshold. The selected samples for all classes are denoted as $\mathbf{I}$

D) determine new pseudo-labels for the unlabeled instances. We use $q(z) = \hat{p}_{f^{(t)}}(z)$ for $z \in \mathbf{I}$ (selected samples in step C)) and

$$v(z) = \frac{1}{2}\Big(\frac{1}{K} + \hat{p}_{f^{(t)}}(z)\Big), \quad \text{for} \ \ z \in \mathbf{U}\backslash\mathbf{I}. \tag{2}$$

E) train a new model $f^{(t+1)}$ by minimizing the loss on labeled and pseudo-labeled samples with AutoAugment (Cubuk et al., 2019) as strong data augmentation:

$$\frac{1}{n + |\mathbf{I}|}\Big[\sum_{i=1}^{n} L\big(y_i, \hat{p}_{f^{(t+1)}}(x_i)\big) + \sum_{z \in \mathbf{I}} L\big(q(z), \hat{p}_{f^{(t+1)}}(z)\big)\Big]$$
$$+ \frac{1}{|\mathbf{U} \setminus \mathbf{I}|}\sum_{z \in \mathbf{U}\backslash\mathbf{I}} L\big(v(z), \hat{p}_{f^{(t+1)}}(z)\big) \tag{3}$$

F) $t \leftarrow t + 1$ and go to step A)

**The Base classifier** is essentially an Outlier Exposure (OE) model (Hendrycks et al., 2019) (see also (Hein et al., 2019; Papadopoulos et al., 2019) for related losses) where the unlabeled set $\mathbf{U}$ is our training out-distribution on which we enforce uniform confidence. OE is known to be one of the best methods for out-of-distribution detection. As in our case, a crucial assumption is that the unlabeled samples are partially task-related, it might appear odd to enforce uniform confidence on all of $\mathbf{U}$. However, we show in Section 3.3 that this asymptotically only leads to a down-weighting of the confidence for task-related samples but preserves the Bayes optimal decision and, in particular, enforces close-to-uniform confidence for all unrelated samples.

**A) Calibration:** while normal neural networks are known to be overconfident on in-(Guo et al., 2017) and out-distribution (Nguyen et al., 2015; Hein et al., 2019), the models resulting from enforcing low confidence on unlabeled points (such as OE) tend to be underconfident on the in-distribution. As we use the predictions of the teacher $f^{(t)}$ as new pseudo-labels for the unlabeled data, we calibrate $f^{(t)}$ by minimizing the expected calibration error using temperature rescaling Guo et al. (2017). Thus the teacher model assigns the correct uncertainty score to its predictions on in-distribution samples which improves pseudo-label quality and and stabilizes the training procedure.

**C) Sample Selection:** The most important problem in self-training is to integrate the right samples into the pseudo-labeled set $\mathbf{I}$. While our out-distribution aware teacher is better at discriminating between the in- and out-distribution based on confidence, there are still many samples with highly confident predictions due to the sheer size of the unlabeled dataset ($\geq 10^6$). Note that we select at most the top-$k$ samples (where $k = 5N(t + 1)/K$), but this might still be too much if not sufficiently many task-related examples of a class exist in the unlabeled dataset. We thus need to determine class-specific confidence thresholds to limit the selection. Class-specific selection has been neglected in the literature but is particularly important in practice as the number of task-related examples in the unlabeled dataset typically varies widely between the different classes.

Using the in-distribution validation set, we define the in-distribution threshold for class $c$ as the smallest predicted probability for class $c$ such that the precision for all images which are above this threshold is greater than or equal to $\alpha$ (binary classification problem: class $c$ versus all other classes). An in-distribution precision threshold is especially important if the classification task contains similar classes, as learning from wrongly labeled in-distribution images (a dog that is classified as a cat in CIFAR10) is likely to hurt predictive performance on the in-distribution task even more than the inclusion of an unrelated out-distribution image.

If there exist less than $k$ samples above the threshold for a particular class, we randomly repeat the accepted samples above the threshold to maintain a class-balanced training scheme. Note that it is

much easier and also more interpretable to fix a precision value rather than the choice of a confidence threshold (in particular if the model is not calibrated) as done in Xie et al. (2020).

**D) Pseudo Labels:** for the original labeled dataset we always use one-hot labels. For unlabeled data points that have been selected in $\mathbf{I}$, we determine pseudo-labels $q$ according to the predicted probability distribution over the classes by the calibrated teacher model. Due to the calibration, this should reflect the "correct" uncertainty about these labels. For all remaining images in our unlabeled dataset $\mathbf{U} \backslash \mathbf{I}$, we use a weak form of knowledge distillation by defining pseudo-labels $v$ as the average of the predicted probability distribution of the teacher model and the uniform distribution, given in (2). This has two reasons: i) a purely uniform distribution on $\mathbf{U} \backslash \mathbf{I}$, which in the first iterations might still contain a lot of task-relevant images, leads to a bias as it does not distinguish between task-relevant and irrelevant images, ii) only using soft-labels from the teacher model leads to overconfident predictions as one can observe in the non-out-distribution aware NSST method. Thus a trade-off between these opposing goals is their average which leads to heavy damping of the confidence (note that the pseudo-labels have a maximal confidence of $\frac{1}{2} + \frac{1}{K}$ on $\mathbf{U} \backslash \mathbf{I}$).

**E) Training:** For the final objective in (3), the selected pseudo-labeled samples in $\mathbf{I}$ and the original samples in $\mathbf{T}$ are assigned the same weight. This is quite aggressive as for iteration $t$ we add up to $5t$ as much pseudo-labeled data as labeled training data. However, this also enables larger performance gains given that the sample selection process is successful. Note that the losses on $\mathbf{I} \cup \mathbf{T}$ and on $\mathbf{U} \backslash \mathbf{I}$ have equal weight as the damping of confidences on $\mathbf{U} \backslash \mathbf{I}$ is crucial for the sample selection process. As pseudo-labels are computed on non-augmented images and we use heavy augmentation during training, our scheme can be regarded as offline consistency learning, where the model is encouraged to replicate the teacher's output independent of the randomly selected augmentation. This allows us to limit the number of passes through the entire unlabeled pool to the number of self-training iterations, do model calibration and results in more stable targets. The entire scheme is repeated multiple times until the validation accuracy starts to degrade or a fixed maximum number of iterations is reached.

**ODST+:** In addition to ODST, we provide the variant ODST+ for scenarios where AI safety is critical. It differs from ODST only in step **C)** where we calculate the final threshold as maximum over ODST's in-distribution threshold and an additional out-distribution threshold. This class-specific out-distribution threshold controls the number of task-irrelevant samples that are falsely added into our pseudo-labeled sample pool $\mathbf{I}$. This is done using an extra out-distribution validation set, i.e. a set of natural images that does not contain any class relevant images (we discuss this choice in Section 4). For each class $c$, we compute the $\alpha$-quantile of the predicted probabilities for class $c$ on the out-distribution images which we define as the out-distribution threshold for class $c$.

## 3.3 BAYESIAN DECISION THEORY OF SELF-TRAINING

In this section, we analyze our iterations in the framework of Bayesian decision theory. We show that the base classifier that enforces uniform confidence on the unlabeled points still leads to optimal decisions on the in-distribution. Moreover, we show that the iterative scheme with soft-labels ultimately reaches the optimal classifier which is Bayes optimal on the in-distribution task and maximally uncertain elsewhere. Proofs can be found in the Appendix A.

In this section we assume that the labeled examples $(x_i, y_i)_{i=1}^n$ are drawn i.i.d. from $\mathrm{p}_{\text{in}}(x, y)$ (joint distribution on $\mathbb{R}^d \times \{1, \dots, K\}$) and the unlabeled data $(z_i)_{i=1}^m$ are drawn i.i.d. from $\mathrm{p}_{\text{all}}$ on $\mathbb{R}^d$. In the open-world setting we think of $\mathrm{p}_{\text{all}}$ as the marginal distribution of a mixture of a very large number of classes (much larger than $K$), including the in-distribution ones (see Appendix) in which case it naturally holds for the marginal distribution $\mathrm{p}_{\text{in}}(x)$ that $\mathrm{p}_{\text{in}}(x) > 0$ implies $\mathrm{p}_{\text{all}}(x) > 0$.

In expectation, the ODST base classifier (Eq. (1)) optimizes (we omit the index 0 in $f^{(0)}$):

$$\mathbb{E}_{(X,Y) \sim \mathrm{p}_{\text{in}}}\Big[ L\big(Y, f(X)\big)\Big] + \mathbb{E}_{Z \sim \mathrm{p}_{\text{all}}}\Big[ L\Big(\frac{1}{K}\mathbf{1}, f(Z)\Big)\Big]. \tag{4}$$

**Lemma 3.1** *Let $\hat{p}(k|x) = \frac{e^{f_k(x)}}{\sum_{l=1}^{K} e^{f_l(x)}}$ then the Bayes optimal prediction for the loss (4) is given for any $x$ with $\mathrm{p}_{all}(x) + \mathrm{p}_{in}(x) > 0$ as*

$$\hat{p}(k|x) = \frac{\mathrm{p}_{in}(k|x)\mathrm{p}_{in}(x) + \frac{1}{K}\mathrm{p}_{all}(x)}{\mathrm{p}_{in}(x) + \mathrm{p}_{all}(x)}, \quad k = 1, \dots, K.$$

We have directly provided the optimal predictive probability distribution instead of expressing it in terms of the classifier $f$. Note that $\hat{p}(k|x)$ is a monotonic transformation of $p_{in}(k|x)$ and thus preserves for each point the ranking of the classes according to $p_{in}(k|x)$ and thus the optimal decision does not change. However, the ordering of the confidence $\max_k p_{in}(k|x)$ across different inputs $x$ is influenced significantly by the ratio of $p_{in}(x)$ to $p_{all}(x)$. Non-task relevant instances where $p_{all}(x)$ is larger than $p_{in}(x)$ are significantly down-weighted and thus will not be selected, whereas if $p_{in}(x)$ is much larger than $p_{all}(x)$ the confidence $\max_k \hat{p}(k|x)$ is almost equal to $\max_k p_{in}(k|x)$. Note that the latter case is in particular true for task-relevant images ($p_{in}(x)$ large) as $p_{all}$ is a much more spread out distribution and thus the density value $p_{all}(x)$ will be small. This justifies our OOD aware base teacher and also our post-training calibration step A) as $\hat{p}$ is under-confident on the in-distribution.

The mathematical treatment of our sample selection strategy is difficult, but it is instructive to check the case where at each iteration $t+1$ we impose on all unlabeled points soft-labels, $\hat{p}_t(k|x)$ defined by the Bayes optimal teacher $f^{(t)}$ at iteration $t$. Then we get the total expected loss at iteration $t+1$:

$$\mathbb{E}_{(X,Y)\sim p_{in}}\big[L\big(Y, f^{(t+1)}(X)\big)\big] + \mathbb{E}_{Z\sim p_{all}}\big[L\big(\hat{p}_t(Z), f^{(t+1)}(Z)\big)\big]. \tag{5}$$

**Lemma 3.2** *The Bayes optimal prediction for* (5) *at iteration $t$ for $t \geq 0$ is given for any $x$ with* $p_{all}(x) + p_{in}(x) > 0$ *and* $k = 1, \ldots, K$ *as*

$$\hat{p}_t(k|x) = p_{in}(k|x) + \Big(\frac{p_{all}(x)}{p_{in}(x) + p_{all}(x)}\Big)^{t+1}\Big(\frac{1}{K} - p_{in}(k|x)\Big).$$

In particular, for any $x$ with $p_{in}(x) + p_{all}(x) > 0$ we get:

$$\lim_{t\to\infty} \hat{p}_t(k|x) = \begin{cases} p_{in}(k|x) & \text{if } p_{in}(x) > 0 \\ \frac{1}{K} & \text{if } p_{in}(x) = 0. \end{cases}$$

This is the perfect out-distribution aware classifier: Bayes optimal for the in-distribution and maximal uncertain on non-task-related regions ($p_{in}(x) = 0$). This justifies our approach from a decision-theoretic perspective. In the finite sample case, neural networks are overconfident on far away regions (Hein et al., 2019) and thus we damp the pseudo-labels on the unlabeled part in step D).

## 4    EVALUATION

First, we evaluate ODST/ODST+ on CIFAR10 in the standard SSL setting with 4k labeled images but now using open world unlabeled data with up to 10M unlabeled images and compare it to existing (open-world) SSL methods. Then in our main evaluation, we use the full training sets of CIFAR10/100 together with the full 80 million unlabeled images from 80MTI and show that we can improve performance in this challenging setting. Moreover, we identify that existing (open-world) SSL methods show a strong distribution shift, which is problematic when using these methods for safety-critical applications.

**Training of ODST and NSST  and (open world) SSL baselines:** For the small scale experiments, we use a WideResNet28x2 for all methods and for the full 80M experiments a standard ResNet50 He et al. (2016) and also the larger PyramidNet272 (Han et al., 2017) with ShakeDrop regularization (Yamada et al., 2019). In each self-training iteration, due to computational restrictions, we use fine-tuning to train the PyramidNets and train all other architectures from scratch. We perform three iterations for the large-scale experiments and five iterations for CIFAR10-4k and always report the iteration with the best in-distribution validation error.

In the sample selection step of iteration $t$, we select the top-$k$ predictions on the full unlabeleld dataset per class as potential candidates for the labeled set, where $k = \frac{5tN}{K}$ ($N$ is the size of the labeled set, $K$ the number of classes). Among these points, we select the ones which are above the in-distribution threshold (NSST  and ODST) respectively in- and out-distribution threshold (NSST+ and ODST+). This difference in sample selection is the only difference to the plus version and we note that NSST  and ODST do not need access to any additional data compared to existing approaches. NSST  also serves as ablation study, as ODST only differs in two aspects: i) the OE base model ii) the train loss (3) instead of (1) (See Appendix B.4 for a detailed discussion).

| | | No SSL | | SSL | | | Open World SSL | | | | |
|---|---|---|---|---|---|---|---|---|---|---|---|
| | | plain | OE | MM | FM | NSST | NSST+ | MTCF | DS3L | ODST | ODST+ |
| 4k L | Acc. ↑ | 86.62 | 84.91 | 81.50 | 89.03 | 88.70 | 87.75 | 91.86 | 78.40 | **93.89** | 93.41 |
| 1M TI | FPR ↓ | 78.48 | 31.17 | 84.16 | 66.71 | 82.20 | 80.60 | 62.17 | 85.51 | **10.51** | 16.71 |
| 4k L | Acc. ↑ | 86.62 | 86.57 | 79.92 | 85.43 | 88.15 | 87.88 | 86.76 | - | 92.14 | **92.21** |
| 10M TI | FPR ↓ | 78.48 | 27.13 | 82.28 | 82.03 | 75.23 | 77.48 | 73.43 | - | 19.33 | **13.78** |
| 4k L | Acc. ↑ | 86.62 | 86.30 | 81.26 | 89.53 | 88.30 | 87.82 | 90.39 | - | **94.31** | 94.31 |
| 1M LSUN | FPR ↓ | 78.48 | 43.98 | 85.27 | 75.50 | 77.47 | 75.73 | 54.08 | - | **23.26** | 25.61 |

Table 1: **CIFAR10-4k:** Results for 4k labeled images and an unlabeled dataset containing 41k CIFAR10 training and 1 million respectively 10 million unlabeled images from 80MTI or 1M LSUN images. MixMatch (MM) and DS3L perform worse than the "plain" baseline (shown in red). ODST+ and ODST outperform all methods in terms of accuracy. For out-of-distribution detection, we report the average false positive rate (FPR) at 95% TPR. ODST+ has a FPR more than 30% better than the closest competitors FixMatch (FM) and MTCF.

As standard SSL baselines, we use the state-of-the-art MixMatch (Berthelot et al., 2019) and Fix-Match (Sohn et al., 2020). For open world SSL we use the recent DS3L (Guo et al., 2020) and MTCF (Yu et al., 2020). We only compare to existing approaches in the small scale setting as DS3L and MTCF do not scale to larger unlabeled datasets due to excessive memory consumption (several terabytes) and MTCF's domain training requiring 5 days for 10M datapoints. Self-training methods are superior in this regard as the number of full passes through the unlabeled data is limited by the number of training iterations and labeling can be parallelized arbitrarily to scale to large datasets.

**Unlabeled dataset:** We use 80 million tiny images Torralba et al. (2008) (80MTI) as unlabeled dataset, which has been created by querying 53,464 different nouns from the wordnet hierarchy. Note that CIFAR10 and CIFAR100 are subsets of 80MTI (Krizhevsky & Hinton, 2009), and we remove (near)-duplicates from 80MTI, see Appendix B.5 for details. For the small scale experiments we use randomly selected subsets of 1M and 10M images and additionally a 1M LSUN subset.

**Evaluation metrics:** In an open-world setting, it is important to not suffer from a distribution shift. Thus, in addition to test accuracy, we evaluate the out-of-distribution (OOD) detection performance by reporting average false positive rate over several OOD datasets: test set of CIFAR100 or CIFAR10, SVHN, LSUN-CR(Yu et al., 2015) with Flowers(Nilsback & Zisserman, 2008) plus Food-101(Bossard et al., 2014) for CIFAR10 and FGVC-Aircraft(Maji et al., 2013) for CIFAR100.

### 4.1 CIFAR10-4κ

In the SSL literature(Berthelot et al., 2019; Yu et al., 2020), it is common to use 4000 labeled CIFAR10 samples as train data and 5000 as validation set. The remaining 41k training images are used as unlabeled data and we mix them with different out-of-distribution datasets to create a challenging open-world scenario: i) 1M respectively ii) 10M unlabeled images from 80MTI or iii) 1M LSUN images. Note that 80MTI contains further task relevant samples for CIFAR10.

The class-specific in-distribution threshold is set to $\alpha = 0.98$ (meaning that we accept maximally 2% false positives per class on the in-distribution validation set) and for ODST+ and NSST+ we use the same value as out-distribution threshold. For the creation of an out-distribution validation set for ODST+ and NSST+, there exist two strategies. The first one is to use an existing dataset with a sufficient variety and remove any potentially task-related samples. For this, we use a subset of 2k CIFAR100 images as validation for the 80MTI experiments and remove the classes "bus" and "pickup-truck" as they can be confused with "car" and "truck" from CIFAR10. The second approach is to manually label a sufficient number of unlabeled samples as out-of-distribution. We simulate this by using a subset of 2k unseen LSUN images in the 1M LSUN experiment.

The results in Table 1 show that our open world setting is challenging due to the low number of task-relevant samples in the unlabeled set (lower bounded by 4.1% respectively 0.41%). MixMatch but also the open world adapted method DS3L perform worse (shown in red) than plain training on the 4k labeled samples. Our ODST and ODST+ outperform all other (open world) SSL methods in terms of test accuracy and significantly improve over the OE model that was used as base teacher (Hendrycks et al., 2019). While for 1M unlabeled data points from 80MTI or LSUN, MTCF and

|  | ResNet50 | | | | | | PyramidNet | | | | |
|---|---|---|---|---|---|---|---|---|---|---|---|
|  | plain | OE | NSST | NSST+ | ODST | ODST+ | plain | OE | NSST | NSST+ | ODST+ |
| Acc. ↑ | 96.11 | 96.81 | 96.86 | 96.93 | 97.98 | **98.07** | 98.49 | 98.28 | 98.24 | 98.13 | **98.57** |
| FPR ↓ | 45.54 | 4.40 | 49.72 | 51.76 | 4.30 | **3.84** | 22.76 | 2.56 | 33.51 | 34.71 | **2.40** |

Table 2: **CIFAR10-50k:** We show test accuracy and FPR@95TPR as out-of-distribution detection performance. ODST+ has the best improvement (1.26%) and final test error (1.93%) for Resnet50 and is the only method which improves for the Pyramid272 architecture by 0.29% with 1.43% test error whereas NSST and NSST+ degrade in test performance.

FixMatch perform well, they show severe performance degradation for 10M data points. We believe that the non-adaptive confidence threshold of FixMatch respectively the Otsu threshold of MTCF are not sufficient for the accurate selection of task-related images. Even though the prediction performance is reasonable for 1M samples, the classifiers show a strong distribution shift as illustrated in Fig. 1 and Fig. 6, where we visualize unlabeled samples which are considered to be task-relevant by the different methods. As the failure of MTCF and FixMatch is not noticeable from test accuracy, we strongly suggest that open world SSL papers should visualize high confidence samples and evaluate OOD detection performance. In Table 1 we therefore report the average FPR over different OOD datasets. ODST and ODST+ have a 30% better FPR than any other SSL method. While ODST+ and ODST show similar empirical performance we demonstrate potential advantages of ODST+ regarding AI safety in Appendix B.6. In Appendix C.1 we provide details over the full run of ODST+ and ODST and report corruption robustness on CIFAR10-C.

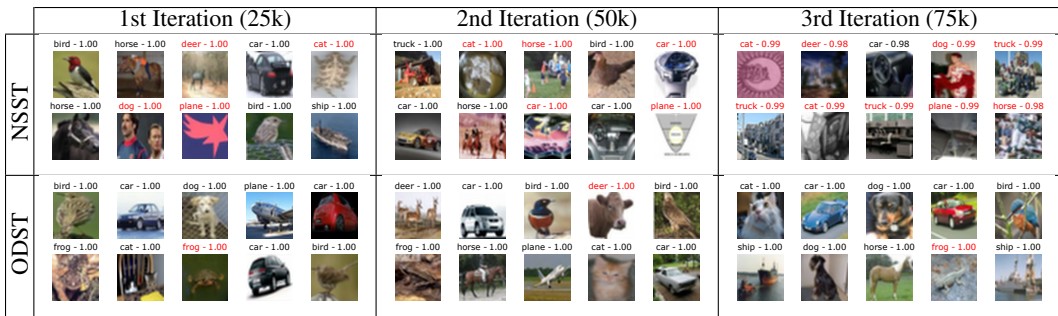

Figure 2: **CIFAR10-50k:** Plot of randomly chosen, exclusively selected samples from 80MTI for NSST (top) and ODST (bottom) over all three iterations. False positives are marked red.

### 4.2 CIFAR10-50k

Up to our knowledge, no SSL method could yet show performance improvements in an open world setting when using the full training set of CIFAR10 as labeled set. We show that this indeed possible when using 80MTI as unlabeled dataset. While DS3L and MTCF do not scale to this setting, both noisy student variants outperform any consistency-based (open world) method in the 10M setting and act as a strong baseline for our ODST and ODST+ models. For the in-distribution validation set, we use the recent CIFAR10.1 dataset (Recht et al., 2018) designed to assess the generalization of classifiers trained on CIFAR10. As out-distribution validation set for ODST+, we use 2k CIFAR100 samples without conflicting classes (See Appendix B.6 for details). As threshold parameter $\alpha$ we use 99.8% which is conservative but justified by the high accuracy of the base CIFAR10 classifiers. In Table 2, we report the results for both architectures.

**ResNet50:** ODST+ improves test accuracy by 1.26% from 96.81% to 98.07% and outperforms NSST and NSST+ by at least 1.14%. We are not aware that such a high test accuracy has been reported before for a ResNet50 on CIFAR10[1]. Even though NSST and NSST+ improve slightly upon the baseline, Figure 2 highlights that even with the OD threshold, NSST+ suffers from a distribution

---

[1] Concerning other results with the same augmentation, Cubuk et al. (2019) require a much larger AmoebaNet-B to achieve an error rate of 1.80% and only achieve 2.6% with a WideResNet-28x10, which typically outperforms a ResNet50 Zagoruyko & Komodakis (2016)

| | ResNet50 | | | | | | PyramidNet | | | | |
|---|---|---|---|---|---|---|---|---|---|---|---|
| | plain | OE | NSST | NSST+ | ODST | ODST+ | plain | OE | NSST | NSST+ | ODST+ |
| Acc. ↑ | 80.69 | 79.98 | 82.44 | 82.03 | 83.54 | **84.09** | 88.07 | 87.60 | 87.71 | 87.53 | **88.66** |
| FPR ↓ | 72.05 | 43.88 | 73.47 | 82.54 | **35.64** | 38.72 | 60.20 | **27.97** | 74.90 | 73.91 | 29.22 |

Table 3: **CIFAR100-45k:** ODST (+) have the best accuracy for both model architectures. SSL-results in red are worse than using labeled data only.

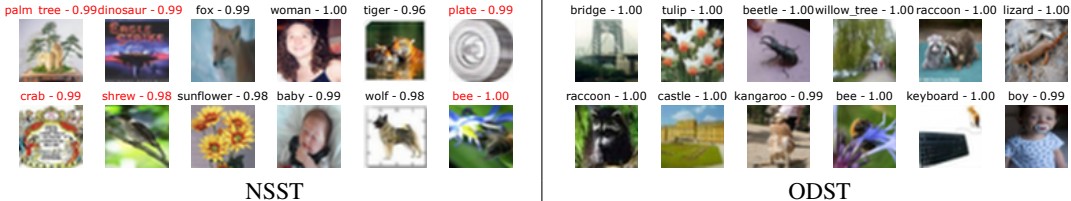

NSST ODST

Figure 3: **CIFAR100-45k:** Random selection of 14 samples in the third iteration of self-training for the ResNet50 architecture. False positives are marked in red.

shift and identifies completely unrelated samples as task-relevant and thus self-training degrades very early e.g. almost all images containing humans are classified as "horse" or "dog". ODST+ slightly outperforms ODST, most likely due to a better sample selection. But even without the additional OOD-validation set ODST improves the OOD performance from $4.40\%$ FPR to $4.30\%$, which is further improved by ODST+ to $3.89\%$. We highlight that all other methods show worse FPR than the base model, which is likely caused by the distribution shift which is concerning regarding AI safety. This again highlights that just relying on test accuracy can be misleading.

**PyramidNet272:** As the test error of the baseline OE is already below $2\%$, further improvements are much harder to realize and can only be obtained by succeeding in the challenging task of selecting high-quality, task-relevant samples from the large pool of unlabeled samples. ODST+ achieves this and is able to improves test accuracy to $98.57\%$ and FPR to $2.40\%$. In contrast, due to their poor sample selection, NSST and NSST+ start to degrade in performance after the first iteration and are thus not able to profit from unlabeled data. We had to skip ODST due to computational constraints.

### 4.3 CIFAR100-45k

For CIFAR100 we randomly select 50 out of the 500 training samples per class as in-distribution validation set. We use 2k CIFAR10 samples without classes "car" and "truck", as they are ambiguous wrt to "pickup-truck", as out-distribution set validation for the Plus methods. In- and out-distribution thresholds are set to $98\%$, due to the lower base accuracy on CIFAR100. The results are in Table 3, where we see that on the ResNet ODST+ improves the test accuracy by $4.11\%$ to $84.09\%$ compared to the OE model and to $83.54\%$ for ODST. While NSST and NSST+ improve by $1.75\%$ to $82.44\%$ respectively $1.34$ to $82.03\%$, we again notice a distribution shift which can be observed from a random selection of samples that pass the threshold and are thus considered task-relevant (Figure 3). This is further verified by the very high FPR of $73.47\%$ respectively $74.19\%$ for OOD detection for the ResNet architecture, whereas both ODST variants improve OOD-performance over the OE baseline. For the PyramidNet, both NSST variants are worse than the plain baseline in accuracy and FPR whereas ODST+ clearly improves test accuracy with a small degradation in FPR.

## 5 Conclusion

We show that ODST can leverage large unlabeled open-world datasets with only a tiny fraction of task-related samples and consistently improves over the baseline and other SSL methods. The resulting classifiers are more accurate, robust and show better out-distribution detection performance. Moreover, we observe that all competing methods suffer from a distribution shift, which is problematic regarding AI safety, whereas ODST and in particular ODST+ shows almost no such degradation.

## Reproducibility Statement

We include our entire codebase in the supplementary material of our submission which allows to easily reproduce our results. This includes indices to any train/validation splits and indices of near-duplicates in 80MTI such that it is possible to run the experiments on the same data that was used in the paper. Due to the stochastic nature of neural network training, results can vary slightly.

## Ethics Statement

There are no conflicts of interest in this work. A potential danger of semi-supervised learning methods, in particular self-training, is that they suffer from a distribution shift. This danger is even more severe in an open-world scenario. As we show in this paper, heavy distribution shifts happen for other SSL methods but our ODST is very robust to it. Nevertheless, we have introduced ODST+ where an extra out-distribution validation set is used to prevent distribution shifts. While in the considered classification problems ODST was already very robust, this might seem unnecessary. But in our opinion this is a too much benchmark-focused point of view. In order to build trustworthy ML systems using open-world SSL, we thus encourage subsequent work to i) check out-of-distribution detection performance, ii) check the selected unlabeled samples, and iii) consider an out-distribution validation set for the construction of the models.

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

- Section D contains ablations studies in the large scale setting with the full 80 tiny images setting.

## A   Proofs for Bayesian Decision Theory of Self-Training

We provide here the missing proofs of Section 3.3 where we have analyzed our iterations of self-training in the framework of Bayesian decision theory.

We repeat the setting and assumptions so that this section is self-contained.

**Assumptions:**   Let $x \in \mathbb{R}^d$ be the input and $y \in \{1, \ldots, K\}$ be the class label. We assume that our labeled examples $(x_i, y_i)_{i=1}^n$ are drawn i.i.d. from $\mathrm{p_{in}}(x, y)$ (joint distribution over inputs and labels on $\mathbb{R}^d \times \{1, \ldots, K\}$) which is our in-distribution. The unlabeled data $(z_i)_{i=1}^m$ is drawn i.i.d. from $\mathrm{p_{all}}$ (distribution on $\mathbb{R}^d$) where we think of $\mathrm{p_{all}}$ in an open world setting as the marginal distribution of a mixture of a very large number of classes (much larger than $K$), including the in-distribution ones. We also assume that $\mathrm{p_{in}}(x) > 0$ implies $\mathrm{p_{all}}(x) > 0$.

**An open world setting:**   We now present a particular way how such a setting could arise even though this is by no means the only setting under which the above assumptions are realized. However, the following is a good model for 80MTI as the datasets CIFAR10 and CIFAR100 have been generated by subsampling from 80MTI and thus the classes of CIFAR10 respectively CIFAR100 are definitely a subset of all classes present in 80MTI. We assume that every class has a specific distribution $p(x|y)$ and for convenience we assume that they are ordered such that the classes $y \in \{1, \ldots, K\}$ are our task-relevant classes and the ones $y \in \{K + 1, \ldots, M\}$ are other classes not part of the task. Then if we assume that there is no distribution shift, that is $p(x|y)$ is the same for labeled and unlabeled data, then for $y \in \{1, \ldots, K\}$

$$\mathrm{p_{in}}(x, y) = p(x|y)\mathrm{p_{in}}(y),$$

where $\mathrm{p_{in}}(y) > 0$ is the class probability in the in-distribution task and

$$\sum_{k=1}^{K} \mathrm{p_{in}}(k) = 1.$$

Note that

$$\mathrm{p_{in}}(x) = \sum_{k=1}^{K} \mathrm{p_{in}}(x, k) = \sum_{k=1}^{K} p(x|k)\mathrm{p_{in}}(k).$$

The out-distribution is a mixture distribution of all possible classes

$$\mathrm{p_{all}}(x) = \sum_{k=1}^{M} p(x|k)\mathrm{p_{all}}(k)$$

where we assume $\mathrm{p_{all}}(y) > 0$ for all $y \in \{1, \ldots, M\}$ (all classes are present in the unlabeled data) and it has to hold $\sum_{k=1}^{M} \mathrm{p_{all}}(k) = 1$. Generally, it holds $\mathrm{p_{in}}(y) \neq \mathrm{p_{all}}(y)$ for $y \in \{1, \ldots, K\}$. With this assumption we immediately see that

$$\mathrm{p_{in}}(x) > 0 \quad \Longrightarrow \quad \mathrm{p_{all}}(x) > 0,$$

as if $\mathrm{p_{in}}(x) > 0$ there exists at least one class $s \in \{1, \ldots, K\}$ such that $p(x|s) > 0$ which implies that

$$\mathrm{p_{all}}(x) \geq p(x|s)\mathrm{p_{all}}(s) > 0,$$

as we assume that $\mathrm{p}_{\mathrm{all}}(y) > 0$ for all $y \in \{1, \ldots, M\}$. In a closed world setting there would be no additional classes in the distribution of the unlabeled data $\mathrm{p}_{\mathrm{all}}$, that is $M = K$, whereas in our open-world setting we have $M \gg K$. However, note that this model is more general than assuming that the unlabeled data is sampled from the marginal distribution $\mathrm{p}_{\mathrm{in}}(x) = \sum_{k=1}^{K} p(x|k)\mathrm{p}_{\mathrm{in}}(k)$ of our joint distribution on inputs and labels, even in a closed-world setting, as our model would allow that the class probabilities between labeled and unlabeled data could differ which is quite realistic in a practical setting.

The ODST+ base classifier, see (1), optimizes

$$\frac{1}{N} \sum_{i=1}^{N} L\big(y_i, \hat{p}_{f^{(0)}}(x_i)\big) + \frac{1}{M} \sum_{j=1}^{M} L\big(\frac{1}{K}\mathbf{1}, \hat{p}_{f^{(0)}}(z_j)\big).$$

which in expectation (for simplicity we omit the index 0 in $f^{(0)}$) yields:

$$\mathbb{E}_{(X,Y)\sim\mathrm{P}_{\mathrm{in}}}\Big[L\big(Y, \hat{p}_f(X)\big)\Big] + \mathbb{E}_{Z\sim\mathrm{P}_{\mathrm{all}}}\Big[L\big(\frac{1}{K}\mathbf{1}, \hat{p}_f(Z)\big)\Big]. \tag{6}$$

Note that in the following we overload $\mathrm{p}_{\mathrm{in}}$ so if we write $(X, Y) \sim \mathrm{p}_{\mathrm{in}}$ then we mean the joint distribution over inputs times labels on $\mathbb{R}^d \times \{1, \ldots, K\}$ and if we write $X \sim \mathrm{p}_{\mathrm{in}}$ then we mean that $X$ is distributed according to the marginal distribution.

The idea of Bayesian decision theory is to characterize the optimal classifier given that one has full knowledge about the problem and one could optimize over all measurable functions. Note that feed-forward neural networks can approximate any function to arbitrary precision (K. Hornik & White, 1989) so that this optimal classifier is in principle achievable. The Bayes optimal prediction $f$ over all possible measurable functions is characterized in the following lemma in Section 3.3.

**Lemma A.1** *Let* $(X, Y) \sim \mathrm{p}_{\mathrm{in}}$ *and* $Z \sim \mathrm{p}_{\mathrm{all}}$, $L(y, f(x))$ *is the cross-entropy loss and let* $\hat{p}(k|x) = \frac{e^{f_k(x)}}{\sum_{l=1}^{K} e^{f_l(x)}}$ *then the Bayes optimal prediction* $f$ *for the objective in* (4) *is given for any* $x$ *with* $\mathrm{p}_{\mathrm{all}}(x) + \mathrm{p}_{\mathrm{in}}(x) > 0$ *as*

$$\hat{p}_f(k|x) = \frac{\mathrm{p}_{\mathrm{in}}(k|x)\mathrm{p}_{\mathrm{in}}(x) + \frac{1}{K}\mathrm{p}_{\mathrm{all}}(x)}{\mathrm{p}_{\mathrm{in}}(x) + \mathrm{p}_{\mathrm{all}}(x)}, \quad k = 1, \ldots, K.$$

**Proof:** We can write the expected loss in (6) with the predictive distribution $\hat{p}_f(k|x) = \frac{e^{f_k(x)}}{\sum_{l=1}^{K} e^{f_l(x)}}$ and the cross-entropy loss, $L(p, \hat{p}) = -\sum_k p_k \log(\hat{p}_k)$, as

$$\mathbb{E}_{(X,Y)\sim\mathrm{P}_{\mathrm{in}}}\Big[L\big(Y, \hat{p}_f(X)\big)\Big] + \mathbb{E}_{Z\sim\mathrm{P}_{\mathrm{all}}}\Big[L\big(\frac{1}{K}\mathbf{1}, \hat{p}_f(Z)\big)\Big]$$

$$= \mathbb{E}_{X\sim\mathrm{P}_{\mathrm{in}}}\Big[\mathbb{E}_Y\big[L\big(Y, \hat{p}_f(X)\big)|X\big]\Big] + \mathbb{E}_{Z\sim\mathrm{P}_{\mathrm{all}}}\Big[L\big(\frac{1}{K}\mathbf{1}, \hat{p}_f(Z)\big)\Big]$$

$$= \int_{\mathbb{R}^d} \mathrm{p}_{\mathrm{in}}(x) \sum_{k=1}^{K} \mathrm{p}_{\mathrm{in}}(k|x) L(e_k, \hat{p}_f(x))dx + \int_{\mathbb{R}^d} \mathrm{p}_{\mathrm{all}}(z) \sum_{k=1}^{K} \frac{1}{K} L(e_k, \hat{p}_f(z))dz$$

$$= \int_{\mathbb{R}^d} L\Big(\sum_{k=1}^{K} \big(\mathrm{p}_{\mathrm{in}}(x)\mathrm{p}_{\mathrm{in}}(k|x) + \frac{1}{K}\mathrm{p}_{\mathrm{all}}(x)\big)e_k, \hat{p}_f(x)\Big) dx$$

$$= \int_{\mathbb{R}^d} \sum_{k=1}^{K} \big(\mathrm{p}_{\mathrm{in}}(x)\mathrm{p}_{\mathrm{in}}(k|x) + \frac{1}{K}\mathrm{p}_{\mathrm{all}}(x)\big) \Big(\log(\sum_{l=1}^{K} e^{f_l(x)}) - f_k(x)\Big)dx$$

where we have used that the cross-entropy loss is linear in the first argument. We optimize pointwise, that is for every $f(x)$, the above objective. As the cross-entropy loss is convex in $f(x)$ in the second argument (note that the log-sum-exp function is convex, see Boyd & Vandenberghe (2004)) we thus get the pointwise objective, $\phi : \mathbb{R}^d \to \mathbb{R}$, for $c = f(x) \in \mathbb{R}^d$ if we we define

$$z_k = \mathrm{p}_{\mathrm{in}}(x)\mathrm{p}_{\mathrm{in}}(k|x) + \frac{1}{K}\mathrm{p}_{\mathrm{all}}(x).$$

as

$$\phi(c) = \sum_{k=1}^{K} z_k \left( \log \left( \sum_{l=1}^{K} e^{c_l} \right) - c_k \right).$$

The optimality condition is then given by

$$\frac{\partial \phi}{\partial c_r}(c) = \sum_{k=1}^{K} z_k \frac{e^{c_r}}{\sum_{l=1}^{K} e^{c_l}} - z_r = 0.$$

which yields

$$\frac{e^{c_r}}{\sum_{l=1}^{K} e^{c_l}} = \frac{z_r}{\sum_{l=1}^{K} z_l}.$$

and thus we get

$$\frac{e^{f_r(x)}}{\sum_{l=1}^{K} e^{f_l(x)}} = \frac{\mathrm{p_{in}}(x)\mathrm{p_{in}}(r|x) + \frac{1}{K}\mathrm{p_{all}}(x)}{\mathrm{p_{in}}(x) + \mathrm{p_{all}}(x)},$$

where we have used that

$$\sum_{l=1}^{K} z_l = \sum_{l=1}^{K} \left( \mathrm{p_{in}}(x)\mathrm{p_{in}}(l|x) + \frac{1}{K}\mathrm{p_{all}}(x) \right) = \mathrm{p_{in}}(x) + \mathrm{p_{all}}(x).$$

Using that

$$\hat{p}(k|x) = \frac{e^{f_k(x)}}{\sum_{l=1}^{K} e^{f_l(x)}},$$

we get the final result. $\qquad \square$

The second result yields the Bayes optimal prediction for an iterated training scheme where the Bayes optimal predictions $\hat{p}_t$ from the previous Lemma A.1 of the Bayes optimal teacher $f^{(t)}$ at iteration $t$ become the soft-labels for the student model $f^{(t+1)}$.

Then we get the total expected loss for the student model $f^{(t+1)}$ at iteration $t + 1$:

$$\mathbb{E}_{(X,Y)\sim\mathrm{P_{in}}} \left[ L\left( Y, f^{(t+1)}(X) \right) \right] + \mathbb{E}_{Z\sim\mathrm{P_{all}}} \left[ L\left( \hat{p}_t(Z), f^{(t+1)}(Z) \right) \right].$$

**Lemma A.2** *The Bayes optimal prediction for* (5) *at iteration $t$ for $t \geq 0$ is given for any $x$ with $\mathrm{p_{all}}(x) + \mathrm{p_{in}}(x) > 0$ and $k = 1, \ldots, K$ as*

$$\hat{p}_t(k|x) = \mathrm{p_{in}}(k|x) + \left( \frac{\mathrm{p_{all}}(x)}{\mathrm{p_{in}}(x) + \mathrm{p_{all}}(x)} \right)^{t+1} \left( \frac{1}{K} - \mathrm{p_{in}}(k|x) \right)$$

**Proof:** We prove this by induction. First we note that for $t = 0$ (base model) we have shown the predictive distribution $\hat{p}_0(k|x)$ in Lemma A.1 to be

$$\hat{p}_0(k|x) = \frac{e^{f_k(x)}}{\sum_{l=1}^{K} e^{f_l(x)}} = \frac{\mathrm{p_{in}}(x)\mathrm{p_{in}}(k|x) + \frac{1}{K}\mathrm{p_{all}}(x)}{\mathrm{p_{in}}(x) + \mathrm{p_{all}}(x)}.$$

Noting that

$$\frac{\mathrm{p_{in}}(x)\mathrm{p_{in}}(k|x) + \frac{1}{K}\mathrm{p_{all}}(x)}{\mathrm{p_{in}}(x) + \mathrm{p_{all}}(x)} = \mathrm{p_{in}}(k|x) + \left( \frac{\mathrm{p_{all}}(x)}{\mathrm{p_{in}}(x) + \mathrm{p_{all}}(x)} \right)^{1} \left( \frac{1}{K} - \mathrm{p_{in}}(k|x) \right).$$

we observe that this agrees with the expression in the lemma to prove. Thus the induction start for $t = 0$ is verified and we go for the induction step. Repeating the derivation of Lemma A.1 we get

$$\hat{p}_{t+1}(k|x) = \frac{\mathrm{p_{in}}(x)\mathrm{p_{in}}(k|x) + \mathrm{p_{all}}(x)\hat{p}_t(k|x)}{\mathrm{p_{in}}(x) + \mathrm{p_{all}}(x)}$$

From plugging in the induction hypothesis we get

$$\frac{p_{in}(x)p_{in}(k|x) + p_{all}(x)\hat{p}_t(k|x)}{p_{in}(x) + p_{all}(x)}$$

$$= \frac{p_{in}(x)p_{in}(k|x)}{p_{in}(x) + p_{all}(x)} + \frac{p_{all}(x)}{p_{in}(x) + p_{all}(x)}\Big[p_{in}(k|x) + \Big(\frac{p_{all}(x)}{p_{in}(x) + p_{all}(x)}\Big)^{t+1}\Big(\frac{1}{K} - p_{in}(k|x)\Big)\Big]$$

$$= \frac{p_{in}(x)p_{in}(k|x) + p_{all}(x)p_{in}(k|x)}{p_{in}(x) + p_{all}(x)} + \Big(\frac{p_{all}(x)}{p_{in}(x) + p_{all}(x)}\Big)^{t+2}\Big(\frac{1}{K} - p_{in}(k|x)\Big)$$

$$= p_{in}(k|x) + \Big(\frac{p_{all}(x)}{p_{in}(x) + p_{all}(x)}\Big)^{t+2}\Big(\frac{1}{K} - p_{in}(k|x)\Big)$$

which finishes the proof. $\square$

In particular, for any $x$ with $p_{in}(x) + p_{all}(x) > 0$ we get :

$$\lim_{t \to \infty} \hat{p}_t(k|x) = \begin{cases} p_{in}(k|x) & \text{if } p_{in}(x) > 0 \\ \frac{1}{K} & \text{if } p_{in}(x) = 0. \end{cases}$$

Note that this is the perfect out-distribution aware classifier: Bayes optimal for the in-distribution and maximal uncertainty on all non-task-related regions ($p_{in}(x) = 0$).

## B    Implementation and Dataset details

As network architecture, we use a WideResNet28x2 for CIFAR10-4k for all methods. The large scale experiments on the entire CIFAR10/CIFAR100 training set with 80MTI as unlabeled data use a ResNet50 respectively a PyramidNet272.

In this section, we present the hyperparameters used to train our ODST+ and the noisy student selftraining models. Note that we use the same set of hyperparameters for ODST+ and the two baselines NSST  and NSST+ for all experiments.

### B.1    WideResNet28x2

For the base model, the only labeled data are the 4k train images. As this yields short epochs, we train the initial model for 5000 epochs with a piecewise schedule and the learning rate decays by a factor 10 at epochs 2000, 3000 and 4000. The initial learning rate is set to 0.1. Even with strong AutoAugment (Cubuk et al., 2019) augmentation (which includes Cutout (DeVries & Taylor, 2017)), due to the small dataset size, the model tends to overfit in the later stages of training. We thus use early stopping on the validation set. In practice, the model with best validation accuracy is often achieved right after the first LR drop at epoch 2000. As we add up to fives times as much unlabeled samples to the "labeled" pool the epochs get significantly longer and thus we reduce to a limit of 1000 of epochs with a proportional rescaling of the learning rate schedule.

For NSST  and NSST+ we use a batchsize of 128 and for both ODST variants, we use a combined batch size of 256 with 128 in-distribution samples, this includes the labeled training samples as well as the unlabeled samples selected by the self-training method to be task-relevant, and 128 samples for the second part of the loss corresponding to the non-selected unlabeled samples (which are sampled randomly from this set). We use standard SGD with Nesterov momentum of 0.9 and weight decay set to 0.0005. All CIFAR10-4k experiments use 5 iterations of self training and with 2k to 10k additional samples (increasing by 2k per iteration, which corresponds to 5 times the number of train samples - which is the ratio we fix for all experiments).

### B.2    ResNet50

Our ResNet50 models are always trained for 250 epochs with piecewise learning rate schedule with a starting learning rate of 0.1 and decay it by a factor of 10 at epochs 100, 150 and 200. The choice of the data augmentation, batchsize, the optimizer and weight decay is the same as for the WideResNet.

We evaluate validation set error throughout the last 20% of the epoch and chose the model with the best validation set performance. For the large scale experiments, we do three iterations and add up 5/10/15 times the number of labeled training samples in the first/second/third iteration.

### B.3   Shakedrop PyramidNet272

For the PyramidNet, the base models are trained for 1000 epochs with a cosine schedule and initial learning rate of 0.05. The batch size is set to 64 for NSST and NSST+ and a total batch size of 128 for ODST+ (64 for in-distribution and 64 for the unlabeled pool). We use Nesterov SGD with a weight decay of 0.0001. For the base models we use AutoAugment as data augmentation.

As training large models for 1000 epochs and with up to 16 times the amount of labeled data is expensive, we use fine-tuning to train the later student models for 55 epochs. We thus always initialize the new student with the previous teacher model's weight. Note that although the student model is initialized with the teacher model that was used to label the unlabeled data, the training loss for the pseudo-labeled data is not zero due to Shakedrop Yamada et al. (2019) and data augmentation. Due to large amounts of noise from both heavy data augmentation and Shakedrop, we use a mixed augmentation strategy for fine-tuning. In detail we train each student for $N$ epochs with AutoAugment and a cosine schedule with initial learning rate of 0.05 that decays to 0 after $N$ epochs. We then do a warm restart and train for another $55 - N$ epochs using a cosine schedule starting at learning rate 0.01 and only use random cropping and flipping as data augmentation. As data augmentation becomes less useful with increasing amounts of data, we set $N$ to 50 for the first iteration and decrease it to 40 and 30 for the second and third iteration (student). Batch size and weight decay remain at 64/128 and 0.0001 for fine-tuning. Due to computational reasons we only train the ODST+ model in this large scale scenario.

### B.4   Noisy Student Self Training

To make NSST and NSST+ more comparable to ODST(+), we adjusted few details about the original noisy-student self-training scheme (Xie et al., 2020) and adapted it to our CIFAR setting, as it was orignally proposed in a larger ImageNet setting with JFT-300M as unlabeled distribution. First, we calibrate all models using temperature scaling (Guo et al., 2017) by minimizing the expected calibration error, whereas Xie et al. (2020) use the non-calibrated classifier outputs. Standard models are typically overconfident and the calibration will make them less overconfident, which should result in better SSL performance as the calibrated pseudo-labels are used in the next iteration.

Next, we calculate the in-distribution threshold (for NSST) and the in- and out-distribution thresholds (for NSST+) on the validation set, whereas Xie et al. (2020) use a fixed threshold of 0.3 for ImageNet. As can be seen by the FixMatch results in Section C, a fixed threshold is generally not sufficient for filtering out task-related samples and it is not obvious how one would adapt the ImageNet threshold to CIFAR10. Our new class-specific thresholding makes the NSST and NSST+ baselines more comparable to ODST+ and helps the two baseline methods to select fewer non-task-related samples. To make the training more comparable to the other SSL methods using a WideResNet28x2, we do not use any form of stochastic depth as in Xie et al. (2020), however our strong data augmentation policy still induces a high amount of noise into the training scheme. On top, the PyramidNet272 with ShakeDrop regularization is even noisier than any architecture used in Xie et al. (2020), however NSST and NSST+ are still underperforming in comparison to ODST+, implying that noise alone is not sufficient to make a self-training scheme robust to the open world setting. In summary, NSST and ODST only differ in two aspects, namely the base model, as ODST uses an OE model in comparison to the plain teacher of NSST and in the training loss, as ODST includes the second loss term from equation (3). All other details, including hyperparameters, calibration and threshold selection are chosen to be identical. Thus all performance improvements can be directly related to the base model and the loss function.

### B.5   Duplicate removal

In this section, we explain our approach to duplicate removal of CIFAR test images in the 80MTI dataset. As both CIFAR10 and CIFAR100 are subsets of 80 million tiny images, it is important to remove exact- and near-duplicates of test images from the unlabeled distribution to prevent them from leaking into our train set. However, after duplicate removal, the unlabeled distribution should still contain task-relevant images to use for semi-supervised learning. First, we noticed that the duplicate removal from Hendrycks et al. (2019) did not remove all duplicates from 80MTI, which they use as out-distribution to enforce uniform confidence (Figure 5a). While this should not improve their test accuracy, it might influence out-distribution detection when for example calculating the false posi-

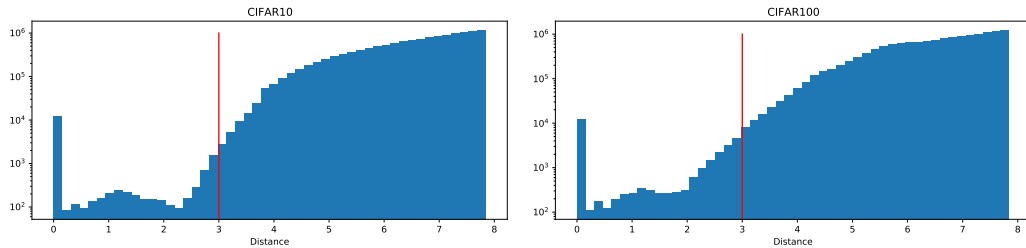

Figure 4: Logarithmic histogram of nearest neighbour distances between CIFAR test set and 80MTI for image pairs with an $l_2$ distance below $2000/255$.

tive rate for the OOD datasets where we use the CIFAR10 and CIFAR100 test sets. Carmon et al. (2019) follow Recht et al. (2018) and remove all 80MTI images with an $l_2$-distance smaller than $2000/255$ to the nearest neighbour in the CIFAR10 test set. While this is likely to remove all duplicates, the approach seems overly strict as after this process, only 65.807.640 out of the 79.302.017 images remain. Thus with their definition of near-duplicate, 80MTI contains nearly 14 million duplicates of the 10.000 test images. In Figure 5b, we show that almost all excluded images are no true duplicates. While we acknowledge that it is important to optimise recall instead of precision when removing duplicates, their procedure is too aggressive and leads to an exclusion of a large set of images which have low variation or close to monochrome images. In Figure 4 we show a histogram of the $l_2$-nearest neighbor distances between the CIFAR test sets and 80MTI. One can see that the vast majority of images have a nearest neighbor distance above an $l_2$-distance of $3.0$. A visual inspection in Figure 5c also confirms that most images below that threshold are duplicates, thus we first remove all images from 80MTI with an $l_2$- distance less than $3.0$ to one of the CIFAR test images. While this removes all exact duplicates, there can exist near-duplicates with larger $l_2$-distance. We thus collect all 80MTI samples with a $l_2$-nearest neighbour to the CIFAR test set smaller than $2000/255$ as potential candidates for removal. For each candidate $x$ and nearest neigbhour $z$ in the CIFAR test sets, we then calculate the perceptual similarity metric LPIPS (Zhang et al., 2018) and SSIM (Wang et al., 2004) and remove the image if $\text{LPIPS}(x, z) < 0.025$ and $1 - \text{SSIM}(x, z) < 0.4$. As both metrics are closer to the visual system, we found them to be more reliable at finding near duplicates for images with larger $l_2$ distances, see Figure 5d, but they are much too expensive to use them directly for nearest neighbor search. We highlight that we do not only find exact duplicates but also degraded versions that for example contain blur, slight translations, color changes and added text or logos. Note that we still remove some non-duplicates, showing that our thresholds are still chosen rather conservatively. Overall, we remove 24k CIFAR10 test set duplicates and 60k CIR100 test set duplicates. Additionally, we also remove all samples selected by Hendrycks et al. (2019), which in particular includes exact train set duplicates. When training CIFAR10 models, we also remove all CIFAR10.1 duplicates with the same approach.

### B.6 ODST+ and the OOD-validation set

In this section, we briefly explain how the OOD-validation set for the plus variants NSST+ and ODST+ can be chosen. This method is especially valuable if the unlabeled pool contains lots of near out-of-distribution samples, for example snakes and lizards can easily be confused with frogs from CIFAR10, especially in a 32x32 resolution. We note that it is impossible to assess the confidence values of a model on near OOD-samples from in-distribution samples, thus one can not easily compute a threshold that allows for a highly precise sample selection without such an OOD-dataset.

As discussed in the main paper, there are two possible ways to generate the OOD-validation set. The first one is to chose a known dataset that covers a large number of classes, for example CIFAR100 or ILSVRC2012 and manually remove the classes that overlap with the train dataset. We use this approach for all experiments with 80MTI as out-distribution. As it is unrealistic to have an OOD-validation set multiple times as big as the train set, we use only 2k CIFAR100 samples in the 4k setting and only use all 50k CIFAR100 images for the PyramidNet in the large scale setting to achieve the best performance possible. In the additional experiments in Appdendix D we further demonstrate that the size of the OOD-validation set does not have a strong influence on the final

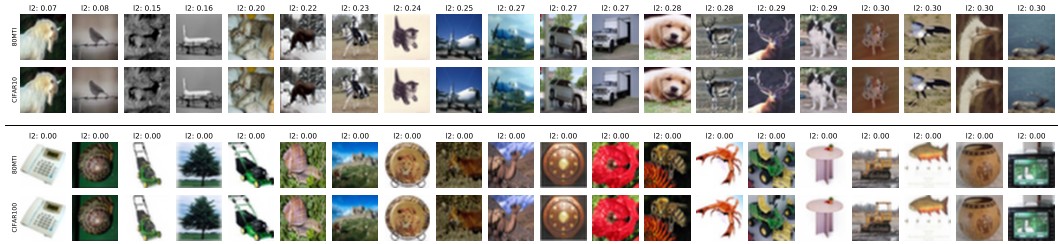

(a) 80MTI images after Hendrycks et al. (2019)'s duplicate removal with their nearest CIFAR10 (upper) and CIFAR100 (bottom) test set neighbours sorted by $l_2$-distance. Note that there are still CIFAR10 test set near-duplicates with distance larger than 0 and exact CIFAR100 test set duplicates.

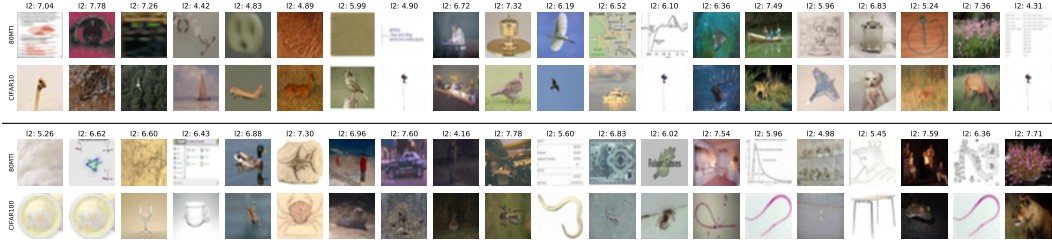

(b) Random selection of 80MTI samples with their respective CIFAR $l_2$-nearest neighbour for sample pairs with distance smaller than $2000/255 \approx 7.84$. This threshold was used by Carmon et al. (2019) for duplicate removal but is too aggressive at it removes too many unrelated images.

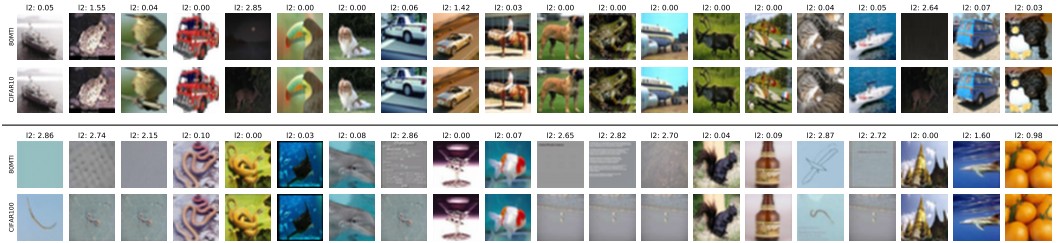

(c) 80MTI samples with their CIFAR $l_2$-nearest neighbour for sample pairs with $l_2$-distance $\leq 3.0$ (random selection). Most pairs are near duplicates and are thus removed in our duplicate removal. Even for this small $l_2$ radius, we find false positives for monochrome images.

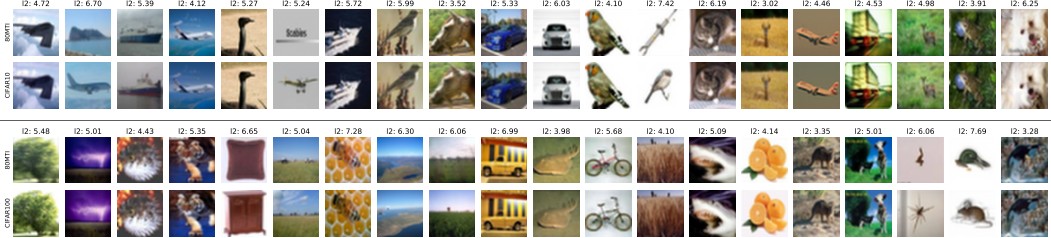

(d) Random selection of samples with $l_2$-distance in $[3, 2000/255]$ that are marked as duplicates wrt to both the LPIPS and SSIM threshold. We are able to find mostly duplicates even in regimes where the $l_2$ distance starts to become meaningless. In total our duplicate removal has a low number of false positives while we could not find any false negative (CIFAR test set image in 80MTI after duplicate removal).

Figure 5: Visualization of various exclusion thresholds for CIFAR10 (above line) and CIFAR100 (below line). The top image shows the sample from 80MTI and the lower one the nearest neighbour in the test set.

quality. For CIFAR100, we similarly use 2k CIFAR10 samples without classes "car" and "truck" and all 50k samples for the PyramidNet experiments.

The second approach for generating such an OOD-validation set is to manually label out-of-distribution samples from the unlabeled pool. This can be done efficiently with a simple interface that sequentially presents samples from the unlabeled pool to the user to let them mark them as task-relevant or out-of-distribution. We simulate this by using 2k unseen LSUN images as OOD-validation set in the 1M LSUN experiments.

## C   Extended results

### C.1   CIFAR10-4k

In this section, we will give some more detailed results on the small scale experiments with only 4000 training images. For MTCF and DS3L we use the official implementations and for FixMatch we use a Pytorch reimplementation[2].

Table 4 contains a breakdown of the 5 iterations making up an entire run for both NSST and ODST. Additionally to the results in the main paper, we also calculate corruption robustness on CIFAR10-C (Hendrycks & Dietterich, 2019). While NSST and NSST+ often decrease in validation accuracy after 2 iterations, ODST is able to increase performance in later iterations and therefore make use of more data. This can be explained by distribution shifts that happen in the first iterations for the NSST models that lead to the addition of too many rubbish samples. In terms of corruption robustness, ODST and ODST+ are able to improve over the OE baseline. In detail, we achieve 78.80% respectively 77.17% for ODST and ODST+ in the 1M TI setting, beating MixMatch (63.41%), FixMatch (77.08%), MTCF (73.08%), DS3L (59.56%), NSST (67.50%) and NSST+ (69.17%). Similarly, we beat the closest competitors MTCF (60.45%), FixMatch (66.19%) and NSST (69.44%) with ODST (74.68%) and ODST+ (74.94%) with 10M TI. With LSUN as out-distribution, we achieve 79.61% and 80.02%, strongly outperforming MixMatch (59.36%), MTCF (72.43%), FixMatch (73.18%), NSST (69.08%) and NSST+ (68.43%).
Overall, ODST and ODST+ are able to clearly outperform all other methods in terms of clean accuracy, corruption robustness and out-of-distribution detection.

We show a large random sample of unlabled images considered to be task relevant by ODST, ODST+, FixMatch and MTCF in Figures 6 and 7 for 1M TI and 1M LSUN images respectively. With 80 million tiny images as out-distribution, both FixMatch and MTCF select too many images that can clearly be labeled as non-task-relevant. For FixMatch, this failure can easily be explained by the fact that they use a fixed confidence threshold of 0.95 on a model that is not trained to explicitly distinguish in- from out-distribution, which results in many OD images with very high confidence values ($> 0.99$). This once again highlights the fact that it is important to be out-distribution-aware during training as well as adapt the confidence threshold to the unlabeled distribution used, for example using an in- and out-distribution threshold like ODST+ does. MTCF is out-distribution-aware as they have an in-versus out-distribution binary classifier which shares the feature representation with the actual classifier. The actual threshold for the binary classifier to select the task-relevant samples from the unlabeled dataset is then calculated via Otsu thresholding. In Figure 6, we show the samples which the final MTCF model considers to be task-relevant when training has finished. In order to understand this more explicitly, we plot the confidence distribution of the CIFAR10 classifier over all unlabeled-samples (left) and of the ones above the Otsu threshold for the ID-vs-OD classifier (right) in Figure 9. It is noticeable that the two classifiers are not perfectly aligned and that relatively low confident samples are able to pass the threshold of the ID-vs-OD classifier whereas a lot of highly confident examples are rejected. The generally lower confidence of MTCF compared to ODST+ and FixMatch seems to be a result of MixMatch training, which MTCF uses for the samples considered to be task-relevant (the pure MixMatch classifier has an even lower average confidence on the in-distribution).

While ODST is not perfect and still includes few unrelated images, it clearly outperforms the other methods by a margin. If an additional OOD-validation set is available, ODST+ can offer a slightly better sample selection quality, but both methods yield strong results given the low amount of labeled training samples together with the low number of task-relevant samples in the unlabeled data.

---

[2]https://github.com/LeeDoYup/FixMatch-pytorch

In Figure 8, for each class, we plot the number of unlabeled samples considered to be task-relevant for ODST+, FixMatch and MTCF. First, we note that all 3 different methods add very distinct class distributions. MTCF for example adds a large number of frogs, however Figure 6 verifies that most of them are indeed false positives. The dominant class for FixMatch is dog, however the samples again show that the method falsely learned to label humans as dogs, resulting in mostly unrelated samples. ODST+ is the most balanced method and has no strong bias towards any class. The peak in cars can be explained by the large amount of car images in 80 Tiny Images and the samples again show that all car samples are indeed cars.

| | ODST | | | | | | ODST+ | | | | | |
|---|---|---|---|---|---|---|---|---|---|---|---|---|
| Iteration | Base | 1st | 2nd | 3rd | 4th | 5th | Base | 1st | 2nd | 3rd | 4th | 5th |
| Validation | 85.22 | 91.44 | 92.52 | 93.74 | 94.12 | **94.62** | 85.22 | 91.50 | 92.70 | 93.62 | 93.68 | **93.96** |
| CIFAR10 | 84.91 | 90.62 | 91.96 | 92.92 | 93.57 | 93.89 | 84.91 | 90.93 | 91.88 | 92.77 | 93.13 | 93.41 |
| CIFAR10-C | 67.19 | 75.43 | 76.32 | 75.47 | 78.38 | 78.80 | 67.19 | 75.29 | 75.48 | 77.65 | 77.05 | 77.17 |
| OOD-FPR95 | 31.17 | 17.24 | 14.04 | 11.95 | 11.34 | 10.51 | 31.17 | 16.84 | 14.51 | 14.38 | 15.45 | 16.71 |

| | NSST | | | | | | NSST+ | | | | | |
|---|---|---|---|---|---|---|---|---|---|---|---|---|
| Iteration | Base | 1st | 2nd | 3rd | 4th | 5th | Base | 1st | 2nd | 3rd | 4th | 5th |
| Validation | 86.50 | 88.06 | 89.04 | **89.20** | 87.08 | 87.92 | 86.50 | 88.62 | 88.62 | **88.64** | 88.34 | 88.28 |
| CIFAR10 | 86.62 | 87.78 | 88.18 | 88.70 | 87.46 | 87.63 | 86.62 | 87.74 | 88.19 | 87.75 | 87.60 | 87.62 |
| CIFAR10-C | 69.04 | 69.65 | 68.69 | 67.50 | 65.46 | 65.40 | 69.04 | 69.24 | 69.38 | 69.17 | 66.71 | 66.38 |
| OOD-FPR95 | 78.48 | 79.10 | 80.13 | 82.20 | 86.71 | 86.57 | 78.48 | 77.45 | 79.75 | 80.60 | 84.50 | 83.71 |

Table 4: **CIFAR10-4k + 1M TI:** Breakdown of the individual iterations of self-training schemes in the CIFAR10-4k setting with an unlabeled pool diluted with 1M samples from 80MTI. The final model is chosen based on validation accuracy and marked in black.

| | ODST | | | | | | ODST+ | | | | | |
|---|---|---|---|---|---|---|---|---|---|---|---|---|
| Iteration | Base | 1st | 2nd | 3rd | 4th | 5th | Base | 1st | 2nd | 3rd | 4th | 5th |
| Validation | 86.78 | 91.40 | 92.98 | 93.00 | **93.28** | 92.88 | 86.78 | 91.72 | 92.46 | **93.06** | 92.90 | 92.76 |
| CIFAR10 | 86.57 | 91.05 | 92.02 | 92.37 | 92.14 | 91.96 | 86.57 | 90.95 | 92.07 | 92.21 | 92.48 | 92.35 |
| CIFAR10-C | 68.10 | 74.21 | 75.35 | 75.40 | 74.68 | 74.22 | 68.10 | 73.88 | 75.14 | 74.94 | 74.43 | 74.26 |
| OOD-FPR95 | 27.13 | 14.29 | 12.68 | 16.97 | 19.33 | 22.52 | 27.13 | 14.06 | 13.19 | 13.78 | 16.98 | 20.52 |

| | NSST | | | | | | NSST+ | | | | | |
|---|---|---|---|---|---|---|---|---|---|---|---|---|
| Iteration | Base | 1st | 2nd | 3rd | 4th | 5th | Base | 1st | 2nd | 3rd | 4th | 5th |
| Validation | 86.50 | 87.92 | **88.74** | 88.16 | 87.74 | 86.78 | 86.50 | 87.52 | **88.64** | 88.62 | 88.16 | 87.56 |
| CIFAR10 | 86.62 | 87.44 | 88.15 | 87.73 | 87.71 | 86.56 | 86.62 | 86.81 | 87.88 | 88.19 | 87.27 | 86.37 |
| CIFAR10-C | 69.04 | 68.65 | 69.44 | 67.91 | 67.30 | 65.20 | 69.04 | 68.50 | 69.08 | 69.25 | 66.33 | 63.80 |
| OOD-FPR95 | 78.48 | 75.76 | 75.23 | 79.67 | 81.03 | 80.91 | 78.48 | 78.92 | 77.48 | 78.52 | 83.47 | 80.54 |

Table 5: **CIFAR10-4k + 10M TI:** Breakdown of the individual iterations of self-training schemes in the CIFAR10-4k setting with an unlabeled pool diluted with 10M samples from 80MTI. The final model is chosen based on validation accuracy and marked in black.

| Iteration | ODST | | | | | | ODST+ | | | | | |
|---|---|---|---|---|---|---|---|---|---|---|---|---|
| | Base | 1st | 2nd | 3rd | 4th | 5th | Base | 1st | 2nd | 3rd | 4th | 5th |
| Validation | 86.80 | 92.16 | 93.82 | 94.38 | 94.98 | **95.04** | 86.80 | 91.94 | 94.06 | 94.54 | 94.82 | **94.88** |
| CIFAR10 | 86.30 | 91.43 | 93.47 | 93.95 | 94.36 | **94.31** | 86.30 | 91.61 | 93.68 | 94.11 | 94.00 | **94.31** |
| CIFAR10-C | 67.70 | 75.67 | 79.74 | 80.12 | 80.14 | **79.61** | 67.70 | 74.95 | 79.07 | 80.09 | 79.67 | **80.02** |
| OOD-FPR95 | 43.98 | 27.98 | 23.42 | 22.83 | 24.55 | **23.26** | 43.98 | 31.49 | 26.17 | 25.16 | 23.03 | **25.61** |

| Iteration | NSST | | | | | | NSST+ | | | | | |
|---|---|---|---|---|---|---|---|---|---|---|---|---|
| | Base | 1st | 2nd | 3rd | 4th | 5th | Base | 1st | 2nd | 3rd | 4th | 5th |
| Validation | 86.50 | 88.32 | **88.76** | 88.04 | 87.54 | 87.92 | 86.50 | 88.52 | **89.32** | 88.54 | 88.50 | 88.52 |
| CIFAR10 | 86.62 | 87.62 | **88.30** | 87.75 | 86.91 | 86.66 | 86.62 | 87.75 | **87.82** | 87.85 | 87.44 | 87.69 |
| CIFAR10-C | 69.04 | 68.66 | **69.08** | 66.82 | 65.02 | 64.43 | 69.04 | 68.48 | **68.43** | 67.41 | 66.84 | 66.79 |
| OOD-FPR95 | 78.48 | 77.61 | **77.47** | 78.73 | 81.58 | 80.43 | 78.48 | 76.47 | **75.73** | 78.22 | 80.36 | 78.95 |

Table 6: **CIFAR10-4k + 1M LSUN:** Breakdown of the individual iterations of self-training schemes in the CIFAR10-4k setting with an unlabeled pool diluted with 1M samples from LSUN. The final model is chosen based on validation accuracy and marked in black.

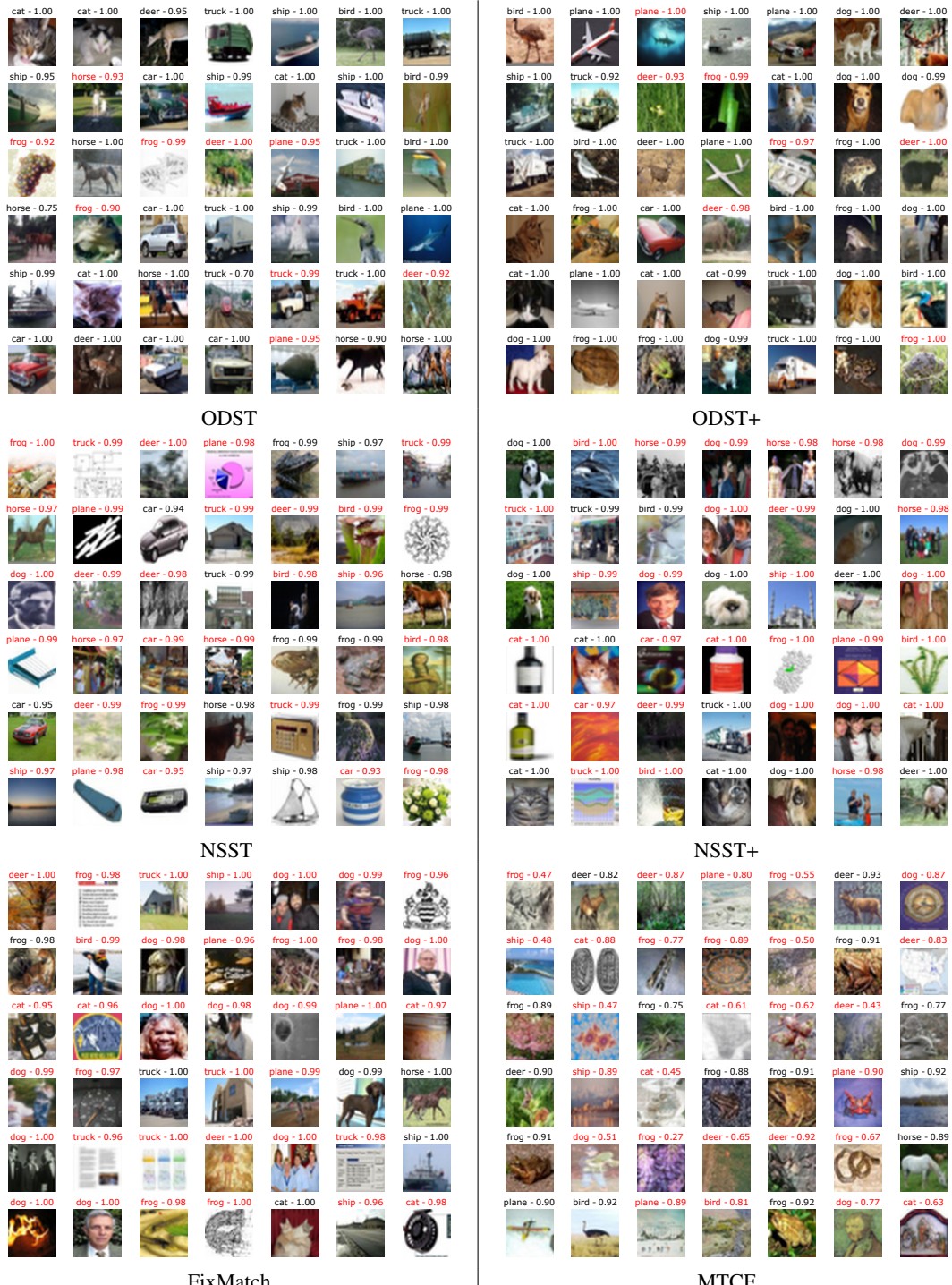

Figure 6: **CIFAR10-4k + 1M TI:** Visualisation of 42 randomly drawn samples considered to be task-relevant for various methods. For ODST and NSST , those are all samples above the in-distribution threshold in the 5th training iteration and above both the in- and out-distribution threshold for ODST+ and NSST+. FixMatch uses a fixed confidence threshold (95%) and MTCF Otsu thresholding to determine which samples are from in- and out-of-distribution. Both variants of ODST are the only methods where almost all selected samples are indeed task relevant and ODST+ achieves slightly higher precision due to the additional OD-threshold. FixMatch and MTCF are able to select task-relevant samples, however both show a high false-positive rate and make systematic errors, for example FixMatch and to some extent MTCF as well learns to label Humans as Dog.

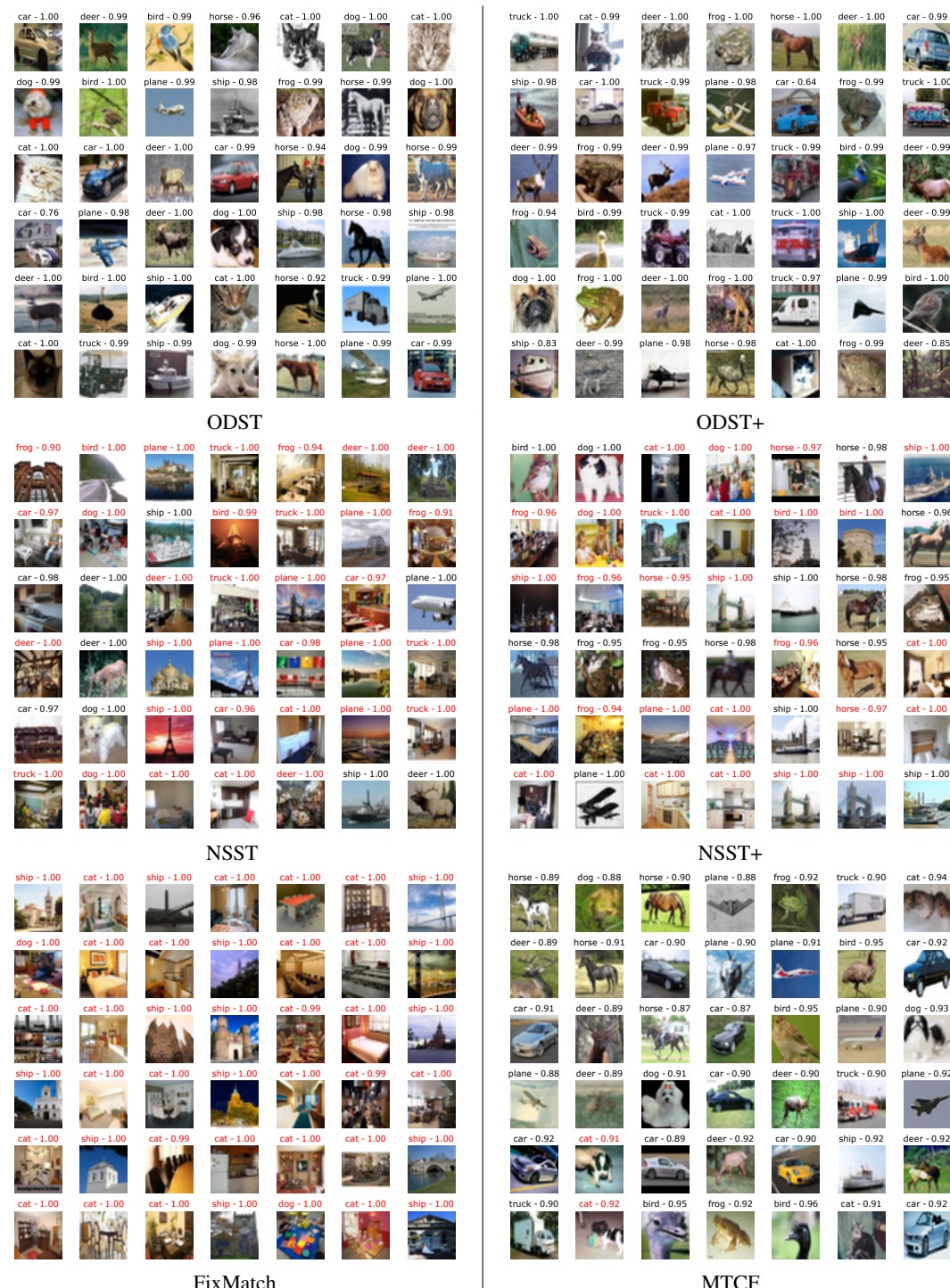

Figure 7: **CIFAR10-4k + 1M LSUN:** Similar to Figure 6, we plot accepted samples for various methods. In this scenario, FixMatch, NSST and NSST+ do once again not show the desired behaviour. On the other hand, ODST, ODST+ and MTCF show nearly perfect sample selection. While MTCF has a great precision (99.70%), we note that it suffers from poor recall (18.47%) and thus does not make use of a majority of available in-distribution samples. ODST and ODST+ on the other hand have a comparable precision (99.26%,98.67%) with a much higher recall (92.81%,92.50%).

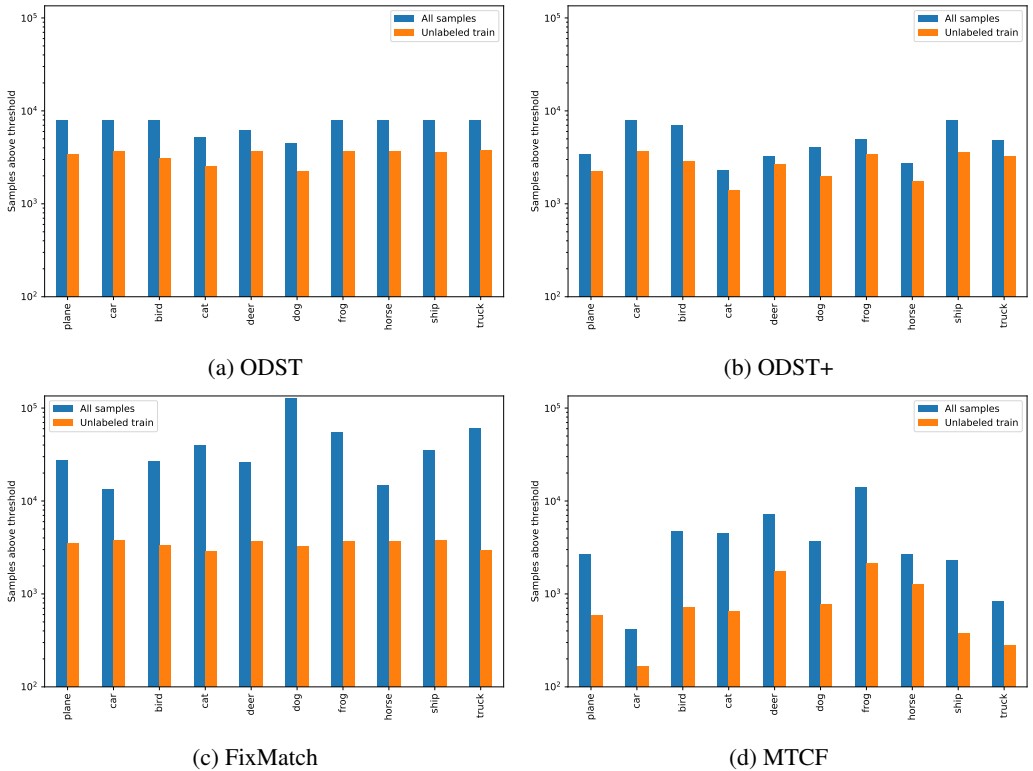

(a) ODST         (b) ODST+

(c) FixMatch         (d) MTCF

Figure 8: **CIFAR10-4k+1M TI:** Number of samples per class from the unlabeled data selected as task-relevant for ODST, ODST+, FixMatch and MTCF with a logarithmic y-axis scaling. The **blue** bar corresponds to the absolute number of added samples and the **orange** bar only shows the samples which come from the unlabeled train data (remember that 41k CIFAR10 samples are in the unlabeled dataset). We note the very uneven selection of MTCF with almost no cars but a lot of frogs (of which most are wrong, see Figure 6), while the selection of ODST+ and FixMatch is more evenly distributed even though FixMatch selects far too many as task-relevant (428k out of 1041k possible). As 80MTI contains some task relevant samples, a perfection selection will have the blue bar slightly higher than the orange one, i.e. it will add the relevant samples out of the 1M tiny images. However, the disparity between the two should not be too large (FixMatch) as this implies that unrelated rubbish images are added. Also, the orange bars should not be too low (MTCF) as this implies that a lot of available in-distribution samples are not used.

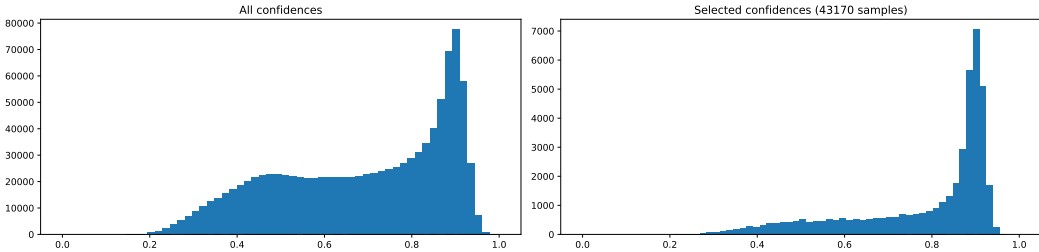

Figure 9: **CIFAR10-4k+1M TI - Analysis of MTCF:** We show the histogram of confidence values of the CIFAR10-classifier of MTCF for all unlabeled samples (**left**) and of the ones which are for the ID-OD-classifier of MTCF above the Otsu-threshold and thus considered task-relevant by MTCF (**right**). Note that MTCF considers a lot of samples task-relevant for which the classifier is quite low-confident and discards the majority of samples for which the classifier is highly confident. This in strong contrast to ODST+ where only unlabeled samples are selected for which the classifier is highly confident (above the in-and out-distribution threshold).

## C.2 CIFAR10-50k

In the CIFAR10-50k setting, we use the full CIFAR10 training set as labeled data and all 80 million tiny images as unlabeled data. For CIFAR10, we use the CIFAR10.1 set of 2k images as validation set. The OOD-validation set for the Plus variants are 2k CIFAR100 samples. In Table 7, we include CIFAR10-C accuracies on top of our standard evaluation metrics test set accuracy and OOD-FPR95 and additionally report which of the three self-training iterations is chosen. Note that our selection criterion is validation accuracy on CIFAR10.1.

On the smaller ResNet50 architecture, ODST and ODST+ can clearly improve performance over the Outlier Exposure (OE) base model in all metrics and also beat the two NSST baselines by a margin. Especially in terms of CIFAR10-C accuracy, we are able to improve from $83.14\%$ achieved by the best labeled-data only model (OE) to $85.09\%$ and $85.97\%$ for ODST and ODST+ respectively. It is noteworthy that both ODST variants are able to improve performance up until the last iteration and therefore make use of the maximum amount of unlabeled data. This is only possible due to a strict sample selection that prevents distribution shifts. NSST suffers from a strong distribution shift in iteration 1 and the resulting model is no longer suitable for passing knowledge to the student model, therefore validation accuracy decreases after the first iteration and we have to select the first iteration model for NSST . NSST+ has the opposite problem and adds too few samples as it is not trained to distinguish in- from out-of-distribution samples but can improve up until the third iteration. However even with the additional threshold, NSST+ is not able to have good precision when it comes to sample selection as can be seen in Figure 10. ODST and ODST+ are the only methods that are able to select mostly task-relevant samples. In this large scale task, ODST+ is also able to beat ODST due to even better sample selection, which is especially important when working with an unlabeled pool that contains 80 million samples.

With the larger PyramidNet architecture, it is even harder to increase performance over the base models (plain and OE) that both already achieve over $98\%$ test set accuracy. Both noisy student variants fail in this setting and are not able to improve performance in any metric and instead show quite significant drops in accuracy and OOD-detection performance over the plain base model. ODST+ is able to improve test accuracy up to $98.57\%$ in the last iteration (starting from $98.28\%$ for the OE teacher model) while not only maintaining but improving FPR.

| | ResNet50 | | | | | | PyramidNet | | | | |
| | plain | OE | NSST | NSST+ | ODST | ODST+ | plain | OE | NSST | NSST+ | ODST+ |
|---|---|---|---|---|---|---|---|---|---|---|---|
| Acc. ↑ | 96.11 | 96.81 | 96.86 | 96.93 | 97.98 | **98.07** | 98.49 | 98.28 | 98.24 | 98.13 | **98.57** |
| C10-C ↑ | 83.03 | 83.14 | 83.59 | 81.02 | 85.09 | **85.95** | **87.79** | 86.40 | 85.28 | 85.97 | 86.95 |
| FPR ↓ | 45.54 | 4.40 | 49.72 | 51.76 | 4.30 | **3.84** | 22.76 | 2.56 | 33.51 | 34.71 | **2.40** |
| Iteration | - | - | 1 | 3 | 3 | 3 | - | - | 1 | 1 | 3 |

Table 7: **CIFAR10-50k:** Additionally to the results from the main paper, we include robustness to common corruptions evaluated on CIFAR10-C (C10-C). The last row "Iteration" refers to which of the 3 self-training iterations is chosen based on validation set accuracy.

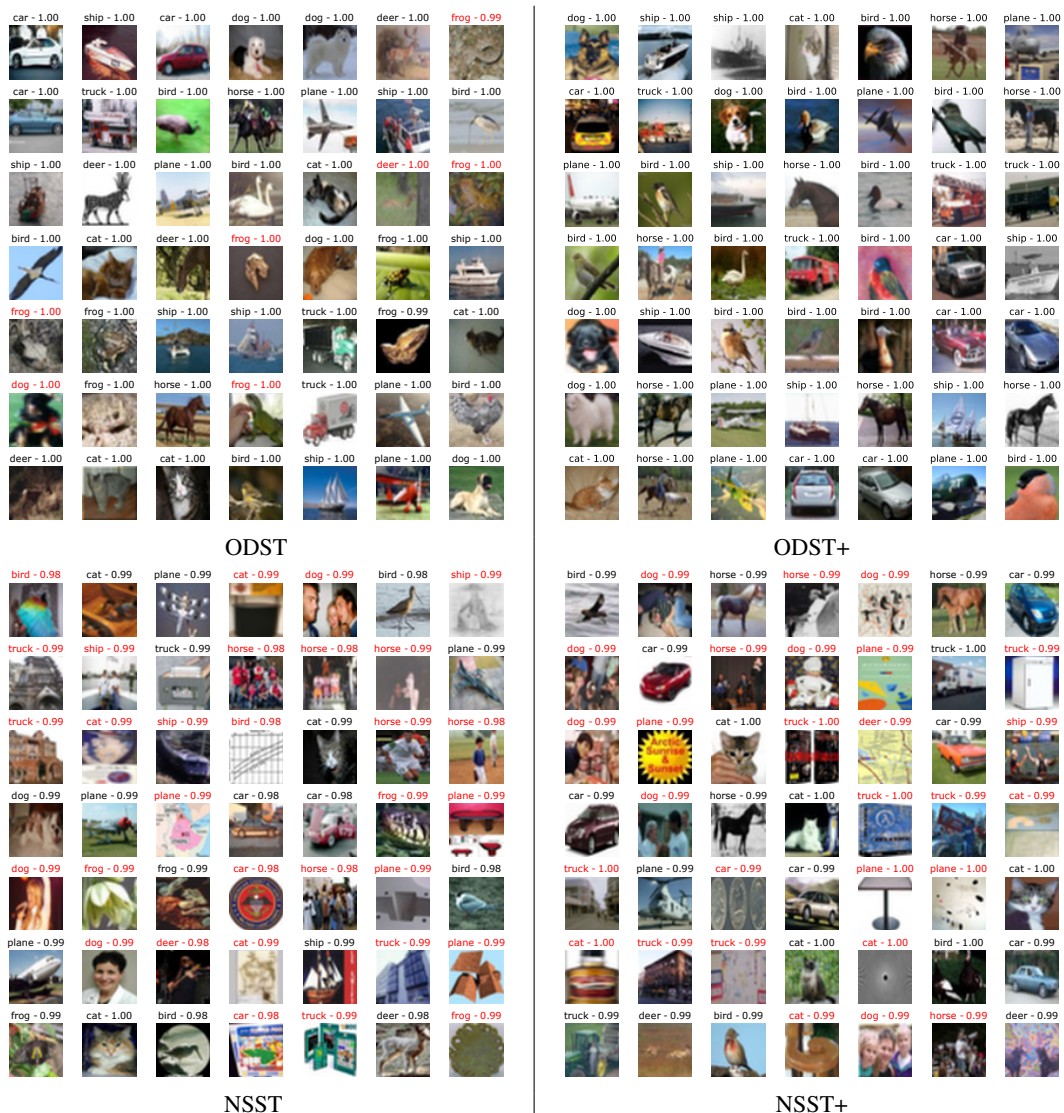

ODST

ODST+

NSST

NSST+

Figure 10: **CIFAR10-50k:** We visualize the selected samples that are over the in-distribution threshold for ODST and NSST and above both the in- and out-distribution threshold for ODST+ in NSST+ in the third iteration of self-training. Note that both NSST variants suffer from a strong distribution shift and start making systematic mistakes, like labeling humans as "dog". Both ODST variants on the other hand are able to select mostly task-relevant samples

## C.3 CIFAR100-45k:

Similar to the CIFAR10-50k setting, we use the full CIFAR100 training set in this setting, however as there is no validation set available we split it into 45k train and 5k validation images. We again use all 80 million tiny images as unlabeled data. The OOD-validation set for the Plus variants are 2k CIFAR10 samples with classes "car" and "truck" removed. In Table 8, we again report corruption robustness measured on CIFAR100-C as well as test set accuracy and OOD-FPR95. Again we also give the chosen self-training iteration based on validation accuracy.

Clearly, ODST+ and ODST outperform all other methods on the smaller ResNet50 architecture. In detail we achieve up to $84.09\%$ test accuracy and $64.02\%$ corruption robustness beating the next best performing methods by a margin ($82.44\%$ and $61.45\%$). In terms of OOD-detection, our FPR values are able to improve significantly over the base outlier exposure model and beat both NSST+ models by more than $30\%$. In Figure 11 we also show a random selection of samples accepted by the various methods and demonstrate thatODST clearly outperforms the NSST baselines in terms of sample selection. The NSST models include many rubbish images that contain diagrams or computer generated graphics that are not even natural images, although the labeled distribution CIFAR100 does not contain such synthetic images. ODST+ and ODST are not perfect, which is to be expected considering that we only see 450 labeled samples per class.

On the larger PyramidNet, ODST+ is the only method that is able to increase test accuracy and achieves $88.66\%$ whereas NSST and NSST+ decrease performance in terms of accuracy, corruption robustness and OOD-detection. ODST+ is able to improve corruption robustness over it's OE base model but beaten by the plain baseline and slightly degrades in OOD detection performance. However the FPR increase from OE to ODST+ is a lot smaller than it is from plain to NSST and NSST+.

| | ResNet50 | | | | | | PyramidNet | | | | |
|---|---|---|---|---|---|---|---|---|---|---|---|
| | plain | OE | NSST | NSST+ | ODST | ODST+ | plain | OE | NSST | NSST+ | ODST+ |
| Acc. ↑ | 80.69 | 79.98 | 82.44 | 82.03 | 83.54 | **84.09** | 88.07 | 87.60 | 87.71 | 87.53 | **88.66** |
| C100-C ↑ | 57.49 | 56.61 | 60.08 | 61.45 | **64.02** | 62.71 | **67.16** | 66.44 | 65.63 | 65.55 | 67.03 |
| FPR ↓ | 72.05 | 43.88 | 73.47 | 82.54 | **35.64** | 38.72 | 60.20 | **27.97** | 74.90 | 73.91 | 29.22 |
| Iteration | - | - | 1 | 1 | 1 | 1 | - | - | 1 | 1 | 2 |

Table 8: **CIFAR100-45k:** Additionally to the results from the main paper, we include robustness to common corruptions evaluated on CIFAR100-C (C100-C). The last row "Iteration" refers to which of the 3 self-training iterations is chosen based on validation set accuracy.

## C.4 SVHN

In addition to the previous CIFAR experiments, we present an additional experiment on the SVHN dataset. Following the CIFAR10-4k setting, we split the original SVHN train set into train/validation/unlabeled splits. We again use 4k labeled train samples, 5k validation samples and the remaining 64k samples as unlabeled data. To simulate an open-world SSL setting, we add 1 million images from 80MTI to the unlabeled set. As baseline methods, we use FixMatch, MixMatch and MTCF and compare it to our ODST. Hyerparameters are the same as in the CIFAR10-4k setting. To evaluate OOD-FPR, we use CIFAR10, CIFAR100, LSUN classrooms and and Flowers.

| | No SSL | | SSL | | Open World SSL | |
|---|---|---|---|---|---|---|
| **SVHN** | plain | OE | MixMatch | FixMatch | MTCF | ODST |
| Acc. ↑ | 94.89 | 94.18 | 93.64 | 96.31 | 96.60 | **96.94** |
| FPR ↓ | 41.64 | 12.20 | 42.31 | 93.89 | **0.1** | 1.1 |

While the task at hand is significantly easier, as distinguishing house numbers from real images is generally a lot easier, ODST is still able to outperform the baseline methods. While FixMatch is able to narrow the gap between ODST and itself in this setting, we want to highlight the catastrophic OOD detection performance that results in an FPR of 94.75.

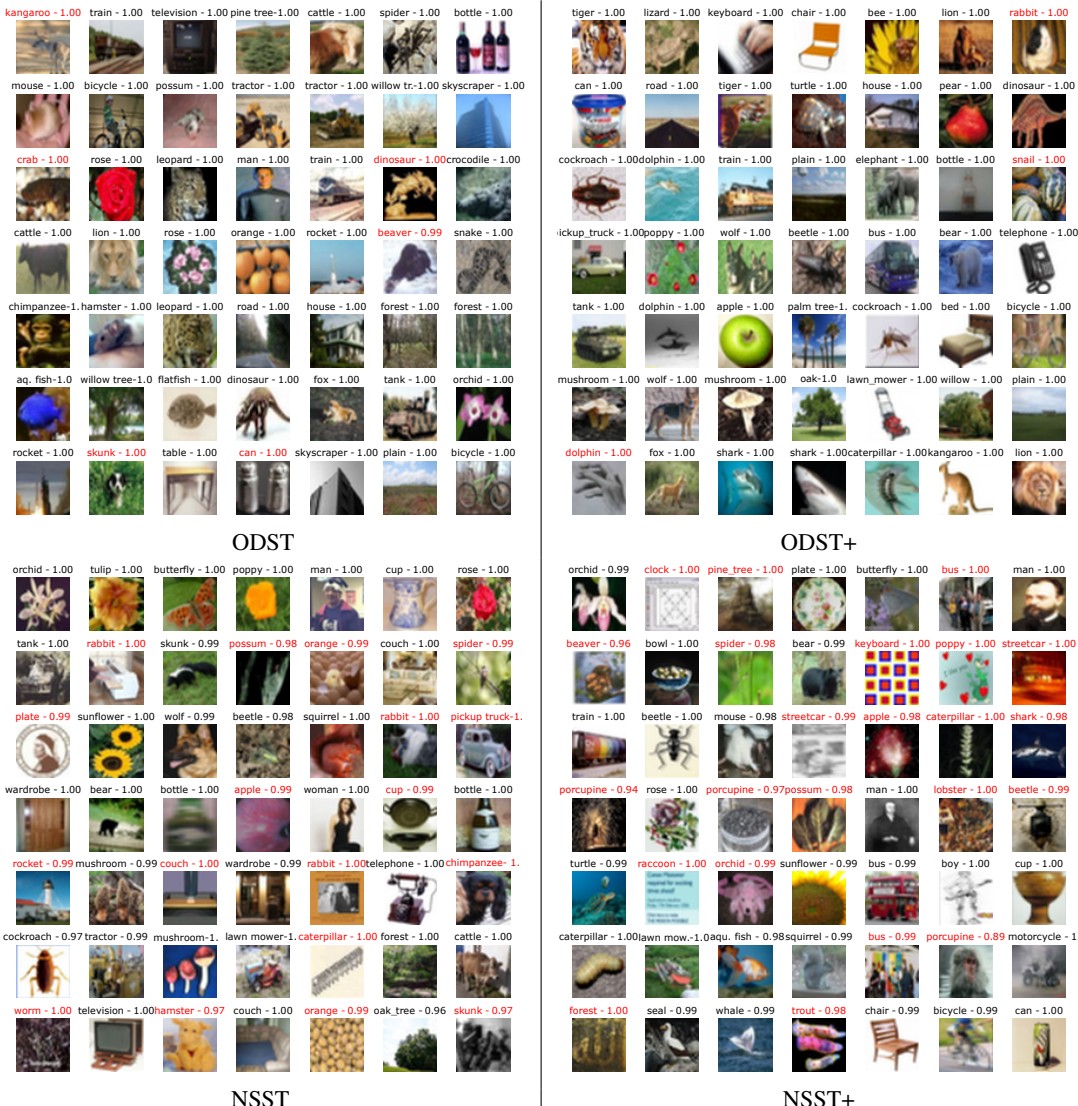

Figure 11: **CIFAR100-45k:** We visualize the selected samples that are over the in-distribution threshold for ODST and NSST and above both the in- and out-distribution threshold for ODST+ AND NSST+ by the first iteration model (Top 50k).

# D  Ablation

In this section, we present various ablation studies to motivate our design choices. All ablation studies use the ResNet50 architecture and are done on CIFAR10-50k with the entire 80 million tiny images as unlabeled distribution.

## D.1  OOD-validation set size

First, in Table 9 we demonstrate that relatively few OOD samples are required to achieve a good performance with ODST+. Both ODST+ variants slightly beat the ODST baseline and perform extremely similar even with a relatiely small OOD-validation set that only contains 2k images. For the rest of this section, we always use the full 50k samples as OOD-validation set.

### D.2 Choice of peudo-labels on $\mathbf{U} \setminus \mathbf{I}$

While Xie et al. (2020) demonstrated that soft-labels on selected samples can improve performance, it remains an open question whether one should use soft-labels for the remaining unlabeled samples in $\mathbf{U} \setminus \mathbf{I}$. In principle, one could keep enforcing uniform confidence on $\mathbf{U} \setminus \mathbf{I}$, that is

$$v(z)_i = \frac{1}{K}, \ i = 1, \ldots, K \text{ for } z \in \mathbf{U} \setminus \mathbf{I},$$

like we did for training the baseline model. Thus instead of training the students with the loss presented in (3), one would minimize the loss:

$$\frac{1}{n + |\mathbf{I}|} \Big[ \sum_{i=1}^{n} L\big(y_i, \hat{p}_{f^{(t+1)}}(x_i)\big) + \sum_{z \in \mathbf{I}} L\big(q(z), \hat{p}_{f^{(t+1)}}(z)\big) \Big]$$
$$+ \frac{1}{|\mathbf{U} \setminus \mathbf{I}|} \sum_{z \in \mathbf{U} \setminus \mathbf{I}} L\big(\mathbf{1}/K, \hat{p}_{f^{(t+1)}}(z)\big). \tag{7}$$

The other alternative is that we use soft-labels on $\mathbf{U} \setminus \mathbf{I}$:

$$v(z) = \hat{p}_{f^{(t)}}(z) \text{ for } z \in \mathbf{U} \setminus \mathbf{I}. \tag{8}$$

In Table 10 we compare these alternative choices to the one of ODST+ (the mean of both) given in Equation (2) which shows that our chosen pseudo-labels in ODST+ are the right compromise between these two extremes.

Using hard-labels on $\mathbf{U} \setminus \mathbf{I}$ decreases performance in comparison to ODST+. There are two possible explanations for this. First, due to our strict thresholding, it is possible that $\mathbf{U} \setminus \mathbf{I}$ contains task-relevant examples that are correctly classified but not accepted into $\mathbf{I}$. In this case, the soft-label is a better target for the next student than strict uniform confidence. Second, even for unrelated images that contain certain features that correlate with a specific class, soft-labels might be a more meaningful target. Soft-labels without label smoothing clearly outperform hard labels in terms of test accuracy but are slightly worse than ODST+.

### D.3 Comparison with 500k-Ti

In this section, we compare our ODST+ scheme with the 500k-Ti dataset Carmon et al. (2019). 500k-Ti was originally designed to help improve adversarial robustness on CIFAR10 by leveraging data from 80MTI. 500k-Ti is generated by training an 11 class classifier and selecting the highest scoring 50k datapoints per-class from the entire 80MTI database. As training data for the 11th background class, Carmon et al. (2019) sample 1M images from 80MTI with keywords not appearing in CIFAR10. Note that their training procedure therefore uses weak labels (the keywords), whereas we do not use this information.

In the experiment, we train a standard classifier by minimizing the cross entropy on the 50k labeled train samples plus 500k-Ti which can thus be seen as the first iteration of a self-training procedure with their selected samples and compare it to ODST+. We use the same hyperpameters as for ODST+, in particular we train a ResNet50 with AutoAugment + Cutout. Table 11 shows that ODST+ clearly outperforms the self-training with 500k-Ti in terms of test accuracy (97.99% vs. 96.82%), corruption robustness (84.98% vs 82.65% and FPR95 for OOD detection 3.25% vs. 39.09%).

| Method | ODST | ODST+ 2k | ODST+ 50k |
|--------|------|----------|-----------|
| Acc. ↑ | 97.98 | 98.07 | **98.12** |
| FPR ↓ | 4.30 | **3.84** | 3.89 |

Table 9: **Ablation:** Influence of the size of the OOD-validation set on test accuracy and out-of-distribution detection performance. ODST does not use any additional OOD-dataset and for ODST+ we use 2k and 50k additional OOD samples from CIFAR100.

| | Base | ODST+ | | | Hard labels $\mathbf{U} \setminus \mathbf{I}$ | | | No label smoothing | | |
|---|---|---|---|---|---|---|---|---|---|---|
| **CIFAR10** | 0th | 1st | 2nd | 3rd | 1st | 2nd | 3rd | 1st | 2nd | 3rd |
| Test accuracy | 96.81 | 97.85 | 97.99 | 98.12 | 97.48 | 97.17 | 97.27 | 97.82 | 97.78 | 97.60 |
| Validation accuracy | 93.00 | 94.60 | 95.45 | 95.30 | 93.60 | 93.35 | 93.40 | 95.10 | 95.25 | 94.60 |
| CIFAR10-C | 83.14 | 85.81 | 84.98 | 84.54 | 84.61 | 85.51 | 86.04 | 85.38 | 84.85 | 84.00 |
| OOD-FPR95 | 4.40 | 3.50 | 3.25 | 3.89 | 3.53 | 3.41 | 3.43 | 3.86 | 5.60 | 8.11 |

Table 10: **Ablation:** Comparison of different choices for the labels of non-selected images in $\mathbf{U} \setminus \mathbf{I}$. ODST+ uses soft-labels with label smoothing factor 0.5, hard labels enforces $1/K$ on all samples and no label-smoothing is equivalent to ODST+ but without any smoothing. All models are trained using the same base model. We again highlight the model with best validation accuracy in black.

| **CIFAR10** | ODST+ | 500k-Ti Carmon et al. (2019) |
|---|---|---|
| Test accuracy | **97.99** | 96.82 |
| CIFAR10-C | **84.98** | 82.65 |
| OOD-FPR95 | **3.25** | 39.09 |

Table 11: **Ablation:** We compare our ODST+ method to a model trained with the 500k-Ti dataset from Carmon et al. (2019). 500k-Ti was created by sampling 50k images per class from 80MTI with the goal of improving adversarial robustness. However, we show that a ResNet50 classifier trained with the same parameters is not able to match the performance of ODST+ in terms of test set accuracy, corruption robustness or out-of-distribution detection.

