# OpenReview forum: "The Needle in the haystack: Out-distribution aware Self-training in an Open-World Setting"
_ICLR.cc/2022/Conference — ICLR 2022 Submitted_

### Official Review · Reviewer_kxNQ · 2021-10-25

**Correctness:** 3
**Technical Novelty And Significance:** 2
**Empirical Novelty And Significance:** 2
**Recommendation:** 5
**Confidence:** 4

**Main Review:**

## strengths
1. This paper studies an important problem which is underexplored in literature;
3. This paper provides analyses on the base classifier based on Bayesian decision theory.
4. The proposed method shows significant performance improvement against several baselines on multiple datasets.


## weaknesses
1. The novelty is limited. The proposed method composes of several parts, i.e., self-training, OE, classifier calibration, confidence-based sample selection, and loss functions are pretty much the same as previous works. which makes the method a combination of existing techniques;
2. The proposed method is complex yet not well understood. More ablation studies are needed to better understand the effectiveness of each component. For example, the last term of the loss function in Eq. (3); the choice of $k$ in sample selection; the classifier calibration; the size of in-distribution validation set;
3. The proposed method is inefficient. In particular, it needs 1000 epochs of training. Additionally, in each epoch, it involves hyper parameter selection on validation set, which adds more computational burden.

**Summary Of The Paper:**

This papers studies semi-supervised learning with OOD samples appearing in unlabeled data as well as at test time, which is practical in real-world tasks. To handle OOD samples, this paper proposes a new framework based on OE and self-trainning. Extensive experiments are conducted to verify the superiority of the proposed method.

**Summary Of The Review:**

As listed in **Main Review**, the novelty seems limited and the proposed method is complex yet not well explained. Therefore, I recommend rejection.

---

> ### Author Response · Authors · 2021-11-23
> **Response to Reviewer kxNQ**
>
> Thank you for taking the time to write your review.
>
> >W1: The novelty is limited. The proposed method composes of several parts, i.e., self-training, OE, classifier calibration, confidence-based sample selection, and loss functions are pretty much the same as previous works. which makes the method a combination of existing techniques;
>
> We would like to note that the OOD-detection based self-training methods are not common in the open-world literature. Indeed, when describing various OOD detection methods “Multi-Task Curriculum Framework for Open-Set SSL” by Yu et al claim that “.., all the methods described earlier need a large number of labeled ID samples to achieve stable results and they are unable to utilize any unlabeled data”. Therefore, most SSL papers that are not in the big data regime (ImageNet as labeled data with hundreds of millions of data points) have focused on consistency-based SSL. In our work, we show that it is indeed possible to train OOD aware models in both a small data (4k labeled samples) and medium data (50k labeled samples) regime and that our self-training based ODST can outperform consistency based SSL methods such as MTCF or FixMatch in both settings.
> However, we also highlight that a simple self-training scheme, such a noisy-student self-training is not able to select the correct samples and significantly improve test accuracy when the unlabeled pool contains a large fraction of out-of-distribution samples.
> While NSST demonstrated that their method is able to improve performance on ImageNet with JFT-300M as unlabeled pool, we demonstrate that this method does not scale to task where the unlabeled pool is too different from the train distribution and suffers from severe distribution shifts and learns wrong class representations (eg classifies humans as dogs), which would prevent the usage of such a model in any practical application.
> Indeed, it is ODST’s unique out-of-distribution aware self-training with an adaptive selection threshold that allows for the largest gains and best OOD performance in a wide variety of settings. This is further verified when we compare ODST to a classifier trained on 500k-TI from “Unlabeled Data Improves Adversarial Robustness” by Carmon et al, which was also created by selecting samples from the entire 80M TI dataset (See Table 11 in the Appendix). Even though 500k-TI has proven to be a valuable tool when it comes to improving adversarial robustness on CIFAR10, a classifier trained on this dataset (and the entire CIFAR10 train set) can not match the performance of ODST(+) in either test accuracy, corruption robustness or out-of-distribution detection.

---

> > ### Author Response · Authors · 2021-11-23
> > **Response to Reviewer kxNQ (continued)**
> >
> > > W2: The proposed method is complex yet not well understood. More ablation studies are needed to better understand the effectiveness of each component. For example, the last term of the loss function in Eq. (3); the choice of k in sample selection; the classifier calibration; the size of in-distribution validation set;
> >
> >
> > We note that the difference between NSST’s and ODST’s training losses is exactly the additional loss term on U\I in Equation (3) (please see answer to Reviewer 3Ko3) - apart from this and the different base classifier ODST and NSST are identical (see page 6 and Appendix B.4 for details on NSST). The removal of this term has the effect that student models become increasingly overconfident on out-of-distribution samples as there is no incentive to predict on OOD-samples with low confidence. This leads then to the addition of task-unrelated samples and causes distribution shifts and wrong class representations as we have demonstrated for NSST. Thus NSST vs ODST is already a main ablation study regarding the last term in Eq. (3). We apologize if this major ablation of NSST vs ODST has not been presented clearly enough.
> >
> > We would like to note that we did not adapt the number of additional samples to our three settings: CIFAR10-4k, CIFAR10-50k and CIFAR100 and demonstrate that our schedule works well with various very different data configurations. This is important, as the number of additional task-related samples in the unlabeled pool is unknown in any practical application so it is very important for the method to work without extensive hyperparameter tuning. However, we also ran an additional experiment for varying the number of maximal additional samples k in the CIFAR10-4k + 1M TI setting (note however that the number of selected samples is influenced in a class-specific way by the in-distribution threshold and for ODST+ also by the in-distribution threshold) and got the following results. Note that we used the shorter training schedule (1000,500,333,250,200 epochs) that achieves an accuracy of 93.60 (see answer W3):
> >
> >
> > | Samples: | 2k/4k/6k/8k/10k  (default) | 3k/6k/9k/12k/15k | 4k/8k/12k/16k/20k |
> > |----------|-----------------|------------------|-------------------|
> > | Accuracy |     93.60       |     93.23        |      93.44        |
> > | OOD-FPR  |     12.11       |     13.86        |      13.52        |
> >
> > While our method is able to outperform all baseline methods with any configuration, adding too many samples in the first few iterations seems to slightly degrade performance over the self-training run.
> >
> > We argue that using calibrated classifiers in a self-training setting is the only reasonable choice as it l directly leads the student classifier to output more reasonable confidence estimates, especially as OE models tend to be underconfident on in-distribution samples. Additionally, we also ran another set of experiments in the CIFAR10-4k + 1M TI setting without calibration (again using the shorter schedule):
> >
> >
> > | Samples: | With Calibration | No Calibration |
> > |----------|------------------|----------------|
> > | Accuracy | 93.60            |  93.37         |
> > | OOD-FPR  | 12.11            |  11.13         |
> >
> > Additionally, our ablation in Appendix D demonstrates that using our smoothed out-of-distribution labels is indeed the right choice and outperforms hard 1/K labels and labels without label smoothing (Table 10).
> >
> > Table 9 in Appendix D shows that the size of the OOD-validation set is indeed not critical for the success of this method. Also, throughout our experiments, we use vastly different sizes for the in-distribution validation set and demonstrate good performance in all of them. For CIFAR10-4k we use 5k validation samples as commonly done in the literature (500 per class). As there is no in-distribution validation set available for CIFAR10-50k we use CIFAR10.1 which only contains 2k samples (200 per class) and for CIFAR100 we use a validation set with 5k samples, which results in only 50 samples per class.

---

> > > ### Author Response · Authors · 2021-11-23
> > > **Response to Reviewer kxNQ (continued)**
> > >
> > > >  W3: The proposed method is inefficient. In particular, it needs 1000 epochs of training. Additionally, in each epoch, it involves hyper parameter selection on validation set, which adds more computational burden.
> > >
> > > First, we would like to stress that the validation set threshold computations are only done once at the beginning of each iteration not at every epoch (please see point D of the algorithmic sketch of ODST on page 4).
> > >
> > > This allows our method to scale to very large datasets with 80M images. This labeling process also only contains forward and no backward passes and can be parallelized to arbitrary GPU configurations if desired. This is in contrast to methods like MTCF, which go over the entire unlabeled dataset 100 times in their domain pretraining, which does not only include forward but also backward passes that are computationally more expensive and have higher memory requirements and prevents them from scaling to task with large unlabeled distributions such as 80M TI.
> > > Also, as the dataset size changes, MTCF can have trouble updating the OOD scores for every sample in the unlabeled pool after domain pretraining. Those OOD scores should reflect the current state of the model, however, they are only updated if those samples are actually selected during training. As the dataset size increases, the probability of selecting a sample decreases which leads to fewer updates and can lead to the fact that precomputed OOD scores are vastly different from what the current model would predict, especially as they use averaging on OOD scores. We believe that this is one of the reasons why MTCF shows severe performance degradations when going from 1M TI to 10M TI in the CIFAR10-4k setting and highlights the advantages of using a fully-trained classifier to predict OOD-scores as ODST does.
> > >
> > > We note that we only use 1000 epochs in the CIFAR10-4k setting and only use 250 epochs in the large-scale CIFAR10-50k setting. Furthermore, we tested a different schedule of 1000, 500, 333, 250 to 200 epochs in the CIFAR10-4k + 1M TI setting and achieve a final accuracy of 93.60%, which clearly outperforms the best baseline MTCF ( 91.86%) while having a similar total computational time (when both methods use a single nVidia V100 gpu). We note that with 10M TI even with 1k epochs, ODST is faster than MTCF due to MTCF doing 100 full passes through all 10 million unlabeled samples.
> > > Also, our PyramidNet experiments in the CIFAR10-50k setting demonstrate that our method is compatible with finetuning which can greatly increase efficiency.

---

### Official Review · Reviewer_EKkN · 2021-11-01

**Correctness:** 2
**Technical Novelty And Significance:** 2
**Empirical Novelty And Significance:** 2
**Recommendation:** 3
**Confidence:** 2

**Main Review:**

This paper focuses on the open-set semi-supervised learning algorithm and proposes a new pseudo-label selection strategy to improve the robustness of SSL algorithms with open-set unlabeled examples.

Strength:
1) The proposal is simple and clear and the paper is well-written.

Weakness:
1) The key technique of the proposal is that the proposal adopts a new pseudo-label threshold to select pseudo labels and the threshold is calculated via in-distribution validation data. In reality, the in-distribution data set is difficult to obtain in a semi-supervised learning setting as we know SSL algorithms are designed to solve the problem with scarce labeled data. The novelty and contribution of the paper are limited.
2) The authors analyze the proposal with bayesian decision theory. The results are mainly based on Eq.(4) and Eq.(5). However, the proposed algorithm does not optimize Eq.(4) and Eq.(5) directly, it seems that the theory can not demonstrate the effectiveness of the proposal.

**Summary Of The Paper:**

This paper focuses on the open-set semi-supervised learning problem. The authors propose a modification of the classical self-learning algorithms. Compared with FixMatch, which selects pseudo-label based on a static confidence threshold. This paper proposes to select a threshold based on in-distribution validation data.

**Summary Of The Review:**

This paper focuses on the open-set semi-supervised learning algorithm and proposes a new pseudo-label selection strategy to improve the robustness of SSL algorithms with open-set unlabeled examples. However, the proposed algorithms are mainly a small modification of current pseudo-label selection-based algorithms, the contribution and novelty of the proposal are limited.

---

> ### Author Response · Authors · 2021-11-23
> **Response to Reviewer EKkN**
>
> Thank you for taking the time to write your review.
>
> >W1: The key technique of the proposal is that the proposal adopts a new pseudo-label threshold to select pseudo labels and the threshold is calculated via in-distribution validation data.
>
> We would like to clarify that our key contribution is not just a class-specific confidence threshold based on the FPR on the in-distribution validation set for the sample selection, but the important key difference to prior work is that we use for self-training an out-distribution aware base model and an out-distribution aware training of the student models (e.g. see response to Reviewer 2). These two contributions together leads to an open-world SSL technique significantly outperforming baselines in various settings on CIFAR10, CIFAR100 and now added also SVHN. We would like to encourage the reviewer to reconsider his/her score as it seems to be based on a major misunderstanding of the contributions of this paper.
>
> > In reality, the in-distribution data set is difficult to obtain in a semi-supervised learning setting as we know SSL algorithms are designed to solve the problem with scarce labeled data. The novelty and contribution of the paper are limited.
>
> The availability of a designated validation set is a common assumption in the SSL literature. However, while we also believe that the split-sizes in the CIFAR10-4k setting of 4k labeled training samples and 5k validation samples that are used in previous works such as Fixmatch or MTCF are rather unrealistic, we use them to stay comparable.
>
> In our own large-scale setting where we use the entire CIFAR10 dataset with 80M TI as unlabeled distribution, we use CIFAR10.1 as validation set which only contains 2k samples, which in our opinion is a lot more reasonable choice in relation to the 50k labeled train samples.
>
> For CIFAR100, we split the 500 train samples per class in a 450/50 train/validation ratio, which results in even fewer validation samples per class than with CIFAR10.1 in the CIFAR10-50k setting (200 per class). Overall, this demonstrates that our method is compatible with vastly different validation set sizes and does indeed not require 500 or more samples per class like in the CIFAR10-4k setting. Note that appendix D1 also evaluates the influence of the OOD-validation set for ODST+ and shows that final performance is only marginally influenced and ODST+ achieves very good results even with a small out-distribution validation set of size 2k.
>
> > W2: The authors analyze the proposal with bayesian decision theory. The results are mainly based on Eq.(4) and Eq.(5). However, the proposed algorithm does not optimize Eq.(4) and Eq.(5) directly, it seems that the theory can not demonstrate the effectiveness of the proposal.
>
> Lemma 3.1 shows that it is indeed reasonable to train the base teacher model, which for all ODST variants is an outlier-exposure like model trained with loss (4), with uniform confidences across all unlabeled samples. This is an important result, as typically outlier exposure assumes that all samples are out-of-distribution. However, in our setting the unlabeled data consists of in- and out-distribution samples and we show that we do not alter the predictions of the Bayes optimal classifier on in-distribution but only the confidence scores. Thus this is directly related to ODST.
>
> For the actual ODST self-training iterations after the training of the initial OE teacher model (Eq 4), we optimize equation (5) but additionally use label-smoothing on out-of-distribution samples. This is due to the fact that we use temperature rescaling on the teacher model which increases confidence outputs as OE models are typically underconfident. As we do not want the model to get overconfident on out-of-distribution samples, we then smooth the predicted probability distributions by averaging them with a uniform distribution. In Appendix D2, we have an ablation study that directly shows the impact of this operation. Especially in terms of out-of-distribution detection performance, we show that the label smoothing version used by ODST improves OOD-FPR95 from 5.60 to 3.25. As being able to detect OOD-samples is highly important in the open-world setting, this also results in better accuracy (97.99 vs 97.78).

---

### Official Review · Reviewer_xS21 · 2021-11-02

**Correctness:** 3
**Technical Novelty And Significance:** 3
**Empirical Novelty And Significance:** 3
**Recommendation:** 5
**Confidence:** 3

**Main Review:**

Strengh: The paper considers a novel mixture model for unlabeled data in an open world scenario. Each unlabeled sample is allocated with a lable with 50% possibility of an outlier, and 50% possibility as an inliner with the psuedo label assigned by a teacher model.

W1: Presentation. I am afraid the presentation quality needs improvement. There are many grammar issues in the paper. Take for example, “A standard setting in SSL is to use 4k training images of CIFAR10 together with 5k as validation set”, is not correct in grammar.

W2: Motivation, I doubt that the pseudo labeling strategy is in fact too noisy, owing to the misalignment of classes between in data and out data. The direct consequence is that the pseudo labeling on out of distribution classes (classes not from the same distribution as from the labeled data) might be seriously wrong. The model would then suffer from the artifact of high confidence (actually, softmax is not capable of measuring any confidence anyway) even if the confidence is already thresholded. If we further take into account the domain gap between in and out data, such plain form of pseudo labeling remains skeptical. The method therefore is sensitive to the threshold selection procedure, subject to the actual domain gap and class difference between in and out data.

W3: Empirical evidence. Due to the lack of presentation clarity, I am curious how the OE model is implemented under a “labeled only” setup as in Table 1, as OE explicitly encourages uncertainty on unlabeled data. Why is OE not compared with the ODST under the same Open World SSL setup? I think OE itself is suitable for such comparisons scenarios.

W4: Theoretical support. The supporting theorem does not link to the propagation error introduced by such pseudo labeling strategy. Since classes between labeled dataset and unlabeled datasets are significantly different, softmax function is prone to prediction artifact. I think what theoretically matters is how much is the error bound constrained in this case given the assumed domain gap and class discrepancy/overlapping between labeled and unlabeled data?

W5: Ablations. There is no ablation study against various components in the proposed ODST method. I insist that the ablation is critical in this paper since the proposed method now is packaged with calibration method, pseudo labeling strategy, student teacher architecture, and the proposed loss Eq. (2). It is currently unclear which factor eventually contributes to the eventual performance improvement.


**Summary Of The Paper:**

The paper assumes an open world setting for semi-supervised learning approaches. In comparison to OE method, the proposed ODST method further dampness the usage of pseudo labels. Specifically, this work further assumes that unlabeled data with lower confidence do come from a mixture model having two components with equal probabilities: one fully uncertain model with label $\frac{1}{K}$ evenly across all classes; and one model with pseudo label determined by the teacher model.



**Summary Of The Review:**

Given the above considerations, I cannot recommend acceptance at this time, unless the above issues are addressed during rebuttal. I would increase my scores if the above concerns are clarified.

---

> ### Author Response · Authors · 2021-11-23
> **Response to Reviewer xS21**
>
> Thank you for taking the time to write your review.
>
> > W1: Presentation. I am afraid the presentation quality needs improvement. There are many grammar issues in the paper. Take for example, “A standard setting in SSL is to use 4k training images of CIFAR10 together with 5k as validation set”, is not correct in grammar.
>
> Thanks for the notice, we will take care of all grammar issues in the final paper and check this with a native speaker.
>
> > W2: Motivation, I doubt that the pseudo labeling strategy is in fact too noisy, owing to the misalignment of classes between in data and out data. The direct consequence is that the pseudo labeling on out of distribution classes (classes not from the same distribution as from the labeled data) might be seriously wrong. The model would then suffer from the artifact of high confidence (actually, softmax is not capable of measuring any confidence anyway) even if the confidence is already thresholded. If we further take into account the domain gap between in and out data, such plain form of pseudo labeling remains skeptical. The method therefore is sensitive to the threshold selection procedure, subject to the actual domain gap and class difference between in and out data.
>
> We fully agree that a standard classifier is not able to reliably assign low confidence (defined as the max over the softmax output) scores to out-of-distribution samples which is well-known in the OOD community. This is exactly the reason why NSST does not work in our open-world setting, where the ratio of non-task-related samples in the unlabeled pool is too large and thus non-task-related samples are selected during self-training. This failure of NSST is exactly the motivation for our ODST method.
>
> This is why we use an Outlier-Exposure (OE) type term on the unlabelled data for the base teacher model. OE reliably assigns low-confidence scores to OOD images, which also results in way better out-of-distribution FPR compared to the plane baseline (CIFAR10-50k: plain: 45.54, OE: 4.40).
> In ODST, we also enforce low-confidence on all unlabeled points, most of them out-of-distribution samples, for all further teacher models during the later iterations of self-training. This is done via the last term in the loss of ODST on the unlabeled point U\I in Equation (3), where we use an average the of the pseudo-labels and a uniform distribution as target distribution. Table 10 in the Appendix shows that this has slight advantages in terms of accuracy for CIFAR10-50k (97.99 vs 97.78) but especially improves OOD-detection capabilities (FPR: 3.25 vs 5.60).
>
> However, out-distribution aware confidence scores alone are not sufficient for good self-training performance and we agree with the reviewer that adaptive sample selection procedures are required as the confidence outputs can vary between datasets, models and in particular across classes. This is why ODST uses a *class-specific* sample selection, instead of fixed confidence thresholds as Fixmatch or Noisy Student, where *class-specific* confidence thresholds are computed by achieving a certain fixed FPR on the in-distribution validation set for the chosen class (as avoiding errors on the in-distribution is important) together with an additional out-distribution threshold for ODST+, which controls the potential amount of selected out-distribution samples. Thus our sample selection procedure is exactly designed to overcome the problems mentioned by the reviewer. Note that we achieve all of this by just fixing FPR values which is more principled than having fixed confidence thresholds as in FixMatch, Noisy Student etc. due to the problems mentioned by the reviewer.

---

> > ### Author Response · Authors · 2021-11-23
> > **Response to Reviewer xS21 (continued)**
> >
> > > W3: Empirical evidence. Due to the lack of presentation clarity, I am curious how the OE model is implemented under a “labeled only” setup as in Table 1, as OE explicitly encourages uncertainty on unlabeled data. Why is OE not compared with the ODST under the same Open World SSL setup? I think OE itself is suitable for such comparisons scenarios.
> >
> > The label "Labeled only’’ in Table 1 is indeed misleading as you correctly point out that OE is trained to enforce low confidence on all unlabeled points. We thus changed the label "Labeled only’’ to "no SSL" but we are open to other suggestions. In Table 10 we present an ablation for the CIFAR10-50k setting where "no label smoothing" corresponds to enforcing  a uniform distribution on the unlabeled points during the self-training iterations as OE would do (everything else is the same as in ODST+) which performs worse than ODST+, in particular the FPR gets worse 5.60% compared to 4.40% of the base model, whereas ODST+ improves this to a FPR of 3.25%. Note also that "no label smoothing" degrades further in the third iteration (FPR of 8.11%) whereas the third iteration of ODST+ improves even in test accuracy and maintains the good FPR of 3.89%. While "no label smoothing" performs still significantly better than NSST+, we think that ODST+ performs better as the "input-agnostic" enforcement of a uniform distribution on the unlabeled points weakens the discrimination of task-related to task-unrelated samples in the unlabeled data leading effectively to a worse selection.
> >
> > > W4: Theoretical support. The supporting theorem does not link to the propagation error introduced by such pseudo labeling strategy. Since classes between labeled dataset and unlabeled datasets are significantly different, softmax function is prone to prediction artifact. I think what theoretically matters is how much is the error bound constrained in this case given the assumed domain gap and class discrepancy/overlapping between labeled and unlabeled data?
> >
> > Our theoretical result shows the effect of an iterated out-distribution aware self-training procedure in the framework of Bayesian decision theory which motivates why self-training can work at all in an open-world setting which is too some extent surprising. The analysis which the reviewer requires would be significantly more complicated and would require a lot of assumptions and, while definitely interesting, is beyond the scope of this paper.
> >
> > > W5: Ablations. There is no ablation study against various components in the proposed ODST method. I insist that the ablation is critical in this paper since the proposed method now is packaged with calibration method, pseudo labeling strategy, student teacher architecture, and the proposed loss Eq. (2). It is currently unclear which factor eventually contributes to the eventual performance improvement.
> >
> > Please note that NSST is already an ablation of ODST as it is already an improvement of the Noisy Student Algorithm of page 3 (please see answer to reviewer 3Ko3 as well), see page 6 and the detailed description of NSST in Appendix B.4. The only difference of NSST and ODST is that ODST uses OE as base classifier compared to a normal classifier for NSST and for all further teacher models ODST uses the average of pseudolabels and uniform distribution as target distribution on the unlabeled samples, whereas NSST has no such term. Everything else: i) calibration of classiifer and the ii) class-specific sample selection is the same for NSST and ODST (showing that NSST is already a significant improvement over the Noisy Studen Paper). This shows that the key technical ingredient which enables open-world self-training is an out-distribution aware teacher model. The removal of the ODST extra terms in the loss in NSST have the effect that later student models become overconfident on out-of-distribution samples as there is no incentive to predict low confidence on OOD-samples. This in return leads to the addition of task-unrelated samples and causes distribution shifts and wrong class representations as we have demonstrated for NSST.
> > We have further ablations as partially discussed above in response to W2 in Appendix D.
> > In terms of calibration, please see our answer to Reviewer kxNQ W2 where we show the behaviour of ODST on CIFAR10-4k without calibration.

---

> > > ### Comment · Reviewer_xS21 · 2021-11-26
> > > **Thanks for your reply**
> > >
> > > Many thanks for the author's kind reply. And thanks for addressing the clarity issues in the empirical study section. I am afraid I am still not convinced by the claims around the effectiveness of the pseudo-labeling strategy though. I am not saying that your proposed theorem is not good. What I am mostly concerned is the way you select the "relevant" samples, since your main novelty is about selecting the samples in an **open world** setting. Due to the possible domain gap and limited generalization ability of the model, your model is actually not capable of dealing with any domain gap (at least in its current formulation) potentially exsisting between your labeled data and unlabeled data. In other words, the selection can be actually very noisy, partly also owing to the artifact out of softmax prediction (since softmax is nothing more than a distance detector that compare the distance between a sample feature against different classes). I therefore think it is more relevant to discuss such issue by associating more theoretical discussions so that we are reassured that domain gap is not a big issue in your challenging **open world** setting, as the open world setup makes domain gap a very relevant issue. I think given the current novelty and the claims, I cannot convince myself that the work has delivered enough evidence which supports the effectiveness of the thresholding strategy both empirically and theoretically in the presence of large domain gap and OOD samples.

---

> > > > ### Author Response · Authors · 2021-11-26
> > > > **Response to Reviewer xS21**
> > > >
> > > > Thank’s a lot for your quick response.
> > > >
> > > > We agree with the reviewer that for a normally trained model the confidence of a classifier computed from the softmax function is not reliable for discriminating between the in- and out-distribution. However, the whole point of the OOD literature in the last years was to show how one can fix this. **Outlier exposure (Hendrycks et al, 2019) has shown that training a classifier by additionally enforcing low confidence on a rich training out-distribution generalizes at test time also to unseen test out distributions in the sense that the confidence of the classifier is a reliable feature to discriminate between in- and out-distribution.** Please see Lemma 3.1 and in particular the discussion of the setting in Appendix A where we explicitly discuss how this works in our open-world SSL setting. While we agree that this is not a full analysis in the framework of learning theory, we don’t know any other paper in the open-world SSL setting which has a theory regarding a potential domain gap. DS3L has a guarantee regarding optimal parameter selection in their algorithm (Theorem 4 in the appendix) but this does not imply any performance guarantee (the expected risk is not upper-bounded in terms of the empirical risk).
> > > >
> > > > **We friendly disagree that we have not delivered enough empirical evidence that for us the selection works fine.** First of all we outperform all other SSL methods (closed/open world) in terms of test accuracy in all tested settings (CIFAR10/CIFAR100/SVHN with less (4k) or more labeled data (full training set)).  We additionally report the OOD-FPR everywhere and outperform all other methods and **we show in the appendix in Figure 6,7,10,11 the selected samples of all methods, which clearly shows that ODST makes the least amount of errors among all competing methods** - this is even true for the challenging CIFAR100 setting where we only have 450 training samples per class (see Figure 11).

---

> > > > > ### Comment · Reviewer_xS21 · 2021-12-03
> > > > > **Thanks for your reply**
> > > > >
> > > > > Thanks for your kind reply. After careful reviews, I think given the current form of presentation along with the novelty, it is still hard for me to recommend the acceptance. Firstly, I did not see any attached proofs in your supplementary material that support the Lemmas. Secondly, Lemma 3.2 seems a bit too magic to me as it guarantees that your pseudo labeling method can 100% accurately recover the OOD sample distribution after many iterations of training. This is very counterintuitive since pseudo labeling is well known to be prone to error propagation and is subject to the domain gaps [1]. Thirdly, the presentation and novelty is a bit concerning to me, for the similar concerns of other reviewers. I guess better interpretation focusing on Lemma 3.2 can definitely help the follow up version, as I think this is the most intriguing part and may form the core contribution of your paper.
> > > > >
> > > > > [1] Shai Ben-David, John Blitzer, Koby Crammer, Alex Kulesza, Fernando Pereira, and Jennifer Wortman Vaughan.
> > > > > A theory of learning from different domains. Machine learning, 2010.

---

### Official Review · Reviewer_3Ko3 · 2021-11-03

**Correctness:** 4
**Technical Novelty And Significance:** 3
**Empirical Novelty And Significance:** 3
**Recommendation:** 8
**Confidence:** 4

**Main Review:**

Semi-supervised learning (SSL) has shown great potential to leverage large amounts of unlabeled data to improve model performance on downstream tasks. However, most of the recent works are hinged on a key assumption that the unlabeled data comes from the same distribution as the underlying task or from the same classes. While some recent works address this by considering the OOD setting, they assume the amount of non-task specific data to be relatively less in the mix. The authors focus on SSL for image classification and consider a more realistic setting where the ratio of non-related images in the unlabeled dataset is large.

To this end, the authors propose a self-training scheme with a careful sample selection strategy. The idea of self-training with unlabeled data has been used in several recent works including Unsupervised Data Augmentation (UDA, NeurIPS 2020) and Noisy Self-training Student (NSST, CVPR 2020) which the authors adopt as the base framework. The authors perform top-K class-based sample selection which has been shown to work well in self-training settings in prior work like Uncertainty-aware Self-training (UST, NeurIPS 2020). They further perform calibration and adapt the self-training loss function to combine (pseduo)-labeled losses from 3 different segments of the data.

The novel component in this method which sets it apart from prior self-training works with sample-selection (including NSST/UDA which uses confidence-based selection and UST which uses uncertainty-based selection) is the use of a weak form of knowledge distillation using pseudo-labels and the uniform distribution. The authors provide theoretical justification for this choice followed by ablation studies in the Appendix to evaluate of such regularization and other components of the framework on the overall model performance.

For the experiments, the authors show impressive improvements over several baselines in both standard SSL and open-world SSL settings.  Comparing NSST+ and ODST+ which is the major component that bring the most benefit for the open-world setting?

Overall, the paper is well-written and easy-to-follow with significant improvements over several baselines in the open-world SSL setting.



**Summary Of The Paper:**

The authors propose an out-of-distribution-aware self-learning (ODST) framework with a careful sample selection strategy that works well with unlabeled datasets containing a small portion of task-relevant data.

**Summary Of The Review:**

The authors propose an out-of-distribution aware self-training method with significant improvements over several baselines in the open-world SSL setting with extensive experiments and ablation studies.

---

> ### Author Response · Authors · 2021-11-23
> **Response to Reviewer 3Ko3**
>
> Thank you for taking the time to write your review.
>
> > Comparing NSST+ and ODST+ which is the major component that bring the most benefit for the open-world setting?
>
> Please note that NSST(+) is already a significantly improved version of the original Noisy Student algorithm provided on page 3 (please see page 6 and the detailed description of NSST in Appendix B.4), as we use the same class-specific confidence thresholding based on the false positive rate on the in-distribution validation set from ODST for NSST, which is already more principled than the fixed confidence threshold used in the original noisy student paper, and we also calibrate the NSST classifiers which is both already an improvement compared to the Noisy Student paper. For the (+) version NSST+ uses then exactly the same thresholding based on the minimum between in-distribution and out-distribution thresholds as ODST+. Thus our NSST in the paper is basically an ablation of ODST where the only difference is that in ODST we explicitly use out-distribution aware teacher models, that is we have an out-distribution aware base teacher model and in each self-training step we enforce a bias towards the uniform distribution on the unlabeled samples as we use as target distribution the average of the pseudo labels and the uniform distribution. This out-distribution awareness of the teacher models in ODST *together* with our principled sample selection is the key to the good performance.

---

> > ### Comment · Reviewer_3Ko3 · 2021-11-27
> > **Response**
> >
> > Thanks to the authors for the clarification. I have read the feedback and all the responses, and maintain my recommendation for the paper.
> >
> > The authors address an important challenge of closed-world SSL setting with their ODST framework with quite extensive experiments,  ablation analysis and significant improvements over baselines that support their claims.
> >
> > Furthermore, the clarifications in all of the reviewer responses can be incorporated to improve the presentation of the work.

---

### Official Review · Reviewer_b8X7 · 2021-11-05

**Correctness:** 4
**Technical Novelty And Significance:** 2
**Empirical Novelty And Significance:** 3
**Recommendation:** 5
**Confidence:** 4

**Main Review:**

I really like the goal of this paper. Most SSL and self-training methods are only evaluated on a contrived dataset where the unlabelled data is actually just labelled data whose labels are removed. For example, a common setup for SSL is to use CIFAR-10, where some images are considered labelled data while rest of the data is considered unlabelled data. The obvious issue here is that the "unlabelled" data in this setup is actually very similar to the labelled data, even with the same distribution of classes etc. So overall I like the goal of this paper.

The method is not very novel. As far as I can tell, the method is similar to noisy student (which itself wasn't too novel or different compared to previous pseudolabelling/self-training methods), with some modifications of calibration, ranking and filtering of confidences, and then relabelling of unlabelled samples with lower confidence pseudolabels. It's worth noting that the original noisy student paper had additional experiments in appendix on filtering psuedolabelled samples by a confidence threshold, where they considered pseudolabels that are hard or soft.

It is of course perfectly fine for a method to not be too novel, as long as it leads to important improvements in empirical results or scientific understanding. I think that the presentation of ODST could be significantly more convincing if it was evaluated on a more challenging benchmark such as imagenet, where noisy student had shown a significant improvement. That said, I do think the results on cifar-4k where ODST outperforms fixmatch is impressive. I'm curious if authors could reference any results of fixmatch applied to task-unrelated unlabelled data from other published papers. Currently as a reader I'm unsure how much of the improvement over fixmatch could be due to fixmatch not being tuned as well as ODST is, since fixmatch has lots of different hyperparameters that could need retuning when unlabelled data is from a task-unrelated source. The authors could also add more information about how they tuned fixmatch and other baseline methods for this harder task where unlabelled samples come from a task-unrelated dataset.

Minor: I see the phrase "resp." on several pages. I wasn't sure if that's a latex bug or not.

**Summary Of The Paper:**

This paper proposes a modified way of doing self-training that aims to improve the performance when the unlabelled data is not coming from task-related labelled data with artificially removed labels. The authors validate their approach on reduced CIFAR-10, full CIFAR-10, and full CIFAR-100.

**Summary Of The Review:**

Since the methodological novelty is limited, the paper could benefit from more comprehensive empirical experiments. Since ODST is quite similar to Noisy-Student, it could be helpful to see experiments on Imagenet. If that is not possible, experiments on more datasets such as STL-10 or SVHN could help, as currently the only experimental evidence for ODST is on CIFAR-10/100.

---

> ### Author Response · Authors · 2021-11-23
> **Response to Reviewer b8X7**
>
> Thank you for taking the time to write your review.
>
> > I really like the goal of this paper. Most SSL and self-training methods are only evaluated on a contrived dataset where the unlabelled data is actually just labelled data whose labels are removed.
>
> Thanks, we fully agree that the evaluation setting of most (open-world) semi-supervised learning methods, where the unlabeled data contains training samples with labels removed is artificial as also criticized by (Olivier et al, NeurIPS 2018). We think that our large-scale open-world setting where the unlabeled distribution is an uncurated dataset containing images from a large variety of classes (such as 80MTI) is more practically relevant as it is unlikely that in practice one has full control over where the unlabeled data is coming from.
>
> > As far as I can tell, the method is similar to noisy student (which itself wasn't too novel or different compared to previous pseudolabelling/self-training methods), with some modifications of calibration, ranking and filtering of confidences, and then relabelling of unlabelled samples with lower confidencepseudolabels.
>
> It is important to note that we compare to an improved version of noisy student training (NSST) which in all experiments only marginally improves over the baseline of using only the labeled data or even degrades performance e.g. for the PyramidNets Table 2 (CIFAR10-50k) and Table 3 (CIFAR100-45k) NSST. The main difference of ODST to NSST is the out-distribution aware training, steps D) and E) of ODST, where we enforce low confidence on the unlabelled data points. This, in combination with calibration and our principled class-specific sample selection (instead of simple confidence thresholding as in Fixmatch or in the original Noisy Student paper) makes it actually possible to apply self-training in an open-world setting. Note that we also improve over the open-world SSL techniques MTCF and DS3L that use schemes which are more complicated. Thus simplicity of our method is actually a plus of ODST. *In general, we show in this paper that with ODST one can improve the already strong classification performance when training on the full CIFAR10/CIFAR100 training set using unlabelled data.* Up to our knowledge this has not been shown before and thus we think that our approach, while maybe not novel in each of its components, constitutes a significant improvement in the field of semi-supervised learning.
>
> > It's worth noting that the original noisy student paper had additional experiments in appendix on filtering psuedolabelled samples by a confidence threshold, where they considered pseudolabels that are hard or soft.
>
> This is true but they have no class-specific thresholding according to a fixed false positive rate but use simple confidence thresholding which ensures no quality control of the added training samples. Please note that we have also an ablation study on the use of hard/soft labels on the additional loss term in Appendix D.
>
> > I think that the presentation of ODST could be significantly more convincing if it was evaluated on a more challenging benchmark such as imagenet, where noisy student had shown a significant improvement.
>
> While we agree that an evaluation on ImageNet would be interesting, we have to highlight the computational costs of NS ImageNet training. The ViT paper “AN IMAGE IS WORTH 16X16 WORDS: TRANSFORMERS FOR IMAGE RECOGNITION AT SCALE” states that on the EfficientNet-L2 architecture, NS training requires 12.3k TPUv3-core-days which is out-of-scope for the computational budget of an academic group, even with a smaller model.

---

> > ### Author Response · Authors · 2021-11-23
> > **Response to Reviewer b8X7 (continued)**
> >
> > > I'm curious if authors could reference any results of fixmatch applied to task-unrelated unlabelled data from other published papers. Currently as a reader I'm unsure how much of the improvement over fixmatch could be due to fixmatch not being tuned as well as ODST is, since fixmatch has lots of different hyperparameters that could need retuning when unlabelled data is from a task-unrelated source.
> >
> > Up to our knowledge, FixMatch is typically not evaluated in open-world SSL papers, the only references we could find is the ICLR 2021 reject “OpenCoS: Contrastive Semi-supervised Learning for Handling Open-set Unlabeled Data” and the recent ICCV2021 “Trash to Treasure: Harvesting OOD Data with Cross-Modal Matching for Open-Set Semi-Supervised Learning” from Huang et al. Note that the second paper again only evaluates the methods in a small scale setting obtained by removing most of the labels from train samples and adding only 10k out-of-distribution samples from various distributions such as LSUN. Both methods largely adopt the same hyperparameters as the original FixMatch paper.
> >
> > For our experiments, we also follow the original hyperparameters. In our opinion, the most important parameter of FixMatch is the confidence threshold that is typically set to 0.95 on CIFAR10 and that could potentially lead to better sample selection in the open-world setting. In Table 5 in the FixMatch paper, they evaluate various confidence thresholds and show a mostly monotonous behavior in terms of error rate when either decreasing or increasing the threshold from the default value of 0.95. We, therefore, ran an additional experiment for the CIFAR10-4k setting and tested 0.9 and 0.97, which are the next best-performing values in the FixMatch evaluation. We found that in terms of accuracy, both perform worse than the original confidence threshold of 0.95:
> >
> >
> >
> > | Treshold | 0.90  | 0.95  | 0.97  |
> > |----------|-------|-------|-------|
> > | Accuracy | 87.77 | 89.03 | 86.12 |
> > | OOD-FPR  | 82.14 | 66.71 | 82.14 |
> >
> >
> >
> >
> > The reason for this is that FixMatch is simply not able to distinguish in- from out-distribution based on the confidence (in fact this is true for all normally trained models and is well-known in the OOD community), independent of the threshold used.
> >
> > > The authors could also add more information about how they tuned fixmatch and other baseline methods for this harder task where unlabelled samples come from a task-unrelated dataset.
> >
> > For all other methods, we kept the hyperparameters as in the original papers. We note that for ODST, we made sure to use similar hyperparameters throughout experiments. For example, we set the number of additional samples in the first epoch to be equal to 5 times the amount of labeled data, in the CIFAR10-4k but also in the CIFAR10-50k and CIFAR100 setting without adjusting this to the dataset size and also use the same data augmentation, learning rates and optimizers.
> >
> > > If that is not possible, experiments on more datasets such as STL-10 or SVHN could help, as currently the only experimental evidence for ODST is on CIFAR-10/100.
> >
> > We have added an experiment on SVHN in Appendix C4 with the same setting as for the CIFAR10-4k experiment which is standard in the SSL community.
> > Again, we split the original SVHN training set into 4k labeled train samples, 5k validation samples and use the remaining images (64K) plus 1M tiny images as unlabeled pool in order to have a realistic open world setting.
> >
> >
> >
> > |   Method | plain | OE    | MixMatch | FixMatch | MTCF  | ODST  |
> > |----------|-------|-------|----------|----------|-------|-------|
> > | Accuracy | 94.89 | 94.18 | 93.64    | 96.31    | 96.60 | 96.94 |
> > | OOD-FPR  | 41.64 | 12.20 | 42.31    | 93.89    | 0.1   | 1.1   |
> >
> > To calculate OOD-FPR scores, we use CIFAR10, CIFAR100, LSUN_CR and Flowers.
> >
> > We note that separating SVHN from 80M TI samples is a much easier task, than separating CIFAR10 from other natural image classes in 80M TI. Therefore closed-world SSL methods like FixMatch perform similar to open-world SSL methods like MTCF and ODST regarding test accuracy. However, when evaluating OOD-FPR, we observe that closed world SSL methods like FixMatch and MixMatch perform quite bad with FPR of more than 40%, whereas both MTCF and ODST achieve less then 1.1%. In total, ODST achieves the highest accuracy with 96.94%.
> >
> > > Minor: I see the phrase "resp." on several pages. I wasn't sure if that's a latex bug or not.
> >
> > We replaced all occurences of "resp." with "respectively".

---

> > > ### Comment · Reviewer_b8X7 · 2021-11-24
> > > **Questions about the SVHN experiment**
> > >
> > > I thank the authors for their helpful responses and additional experiments. Regarding the SVHN experiment, I wonder if mixing the core SVHN samples with tiny images is the right experiment. Why not use the slightly OOD SVHN samples that are outside the core set, which is around 600k samples? I've never done the following, but could it be a more informative experiment to use MNIST as the OOD unlabelled samples?
> > >
> > > Given that FixMatch does almost as well as ODST despite the large difference in OOD-FPR, does that suggest that at least for downstream accuracy OOD-FPR is not a useful metric to monitor?
> > >
> > > Since the SVHN experiment is somewhat inconclusive (FixMatch performs almost as well as ODST), maybe experiments on STL-10 might be more informative? (not suggesting the authors run this in the next 5 days, but perhaps for later submissions if the paper is not accepted or for camera-ready stage if it is accepted).

---

> > > > ### Author Response · Authors · 2021-11-26
> > > > **Response to Reviewer b8X7**
> > > >
> > > > Thank's a lot for your quick response.
> > > >
> > > > We did a large-scale experiment earlier (similar to our CIFAR10-50k), where we use the full SVHN  dataset (73k) as labeled dataset and the extra 531k additional SVHN plus the 80M Tiny Image dataset as unlabeled data. The base model using only labeled data achieves an error of 1.94%  and ODST improves this to 1.36% error whereas training in a supervised fashion on the 73k+521k SVHN data yields 1.24% error, so ODST can nearly match the fully supervised setting. The problem is that MTCF does not scale to such large-scale experiments - thus we reported the 4k setting in the rebuttal as this is feasible for all methods.
> > > >
> > > > If you are 100% sure that you will see only in-distribution inputs at test time, then OOD-FPR is of little interest.  In practice, we would argue that often you cannot control the inputs. In this case, you effectively work in an open-world test setting where the OOD-FPR is essential as otherwise, the classifier will not recognize when it is operating "out of its specification" which could have fatal consequences e.g. in autonomous driving or other safety-critical applications. In numbers: in the SVHN-4k experiment for FixMatch, 88.5% out of the 1M tiny images in the unlabeled dataset, e.g. humans, cars, animals, etc, are above the confidence threshold (0.95) and are thus treated as "SVHN"-images by FixMatch. In contrast, ODST only selects 1.6% of the 1M images from the 80M Tiny Images dataset and of these ones at least a larger fraction shows digits or at least structures that are similar to digits. In this setting where the in-distribution SVHN is "far away’’ from the 1M tiny images in the unlabeled data, the consistency regularization on wrongly selected OOD samples seems to have little effect on in-distribution predictions. However, in the CIFAR10-4k experiment, where the 80M Tiny images in the unlabelled data are semantically similar, consistency regularization on wrongly selected OOD samples has a clear negative bias. This is particularly clear in the setting CIFAR10-4k with 10M images added from the 80M TI dataset, where FixMatch performs worse than just training on the labeled data only.
> > > >
> > > > We don’t think that the experiments are inconclusive regarding FixMatch as in open-world scenarios where, in contrast to SVHN, in- and out-distribution are not clearly separated, e.g. for CIFAR10 or CIFAR100, FixMatch is significantly outperformed by ODST (Table 1). You might have missed that for CIFAR10 we also did the 4k setting where we mixed the unlabeled CIFAR10 training images with 1M images from LSUN (Table 1). There ODST outperforms Fixmatch by 4.78% (94.23% vs. 89.53%). That said we are happy to add STL-10 for the camera-ready version but we will not manage to do that in the remaining rebuttal time.

---

### Decision · Program_Chairs · 2022-01-20

**Decision:**

Reject

**Comment:**

This paper proposes a method for self-training in an open-world setting where a significant portion of unlabeled data might include examples that are not task related. The proposed method (ODST) uses a more accurate OOD detection technique which allows an improved sample selection leading to higher accuracy.

Strong Points:
- This paper studies a very important and impactful problem.
- The paper is well-written.
- The empirical results show that the proposed method improves over prior work.
- To better understand the iterative scheme, authors provide theoretical analysis using Bayesian decision theory.

Weak Points:
- Novelty: Given prior work on different variants of noisy students, this work has limited novelty.
- Dataset diversity: The main results are provided for CIFAR-10 and CIFAR-100 datasets which are very similar to each other. During the discussion period, authors added results on SVHN datasets but the accuracy gap between the proposed method and FixMatch is insignificant (FPR gap is higher but since the main goal is improving performance, I think showing accuracy is a more important measure here).
- Connecting theoretical results to the rest of the paper: The paper can be improved significantly if the theoretical results are more connected to the rest of the paper and in particular with the proposed algorithm.

While 4 out of 5 reviewers are recommending rejection, I think this was a very close decision. Most reviewers were concerned with novelty which I think is a valid point. Given that and the fact that the theoretical results are very limited, showing strong empirical results are required to accept this paper. Even though the provided results on CIFAR datasets are strong, the result on SVHN does not show a significant improvement. I understand that running experiments on ImageNet might not be budget-friendly. However, it is possible to run similar experiments or other datasets to show the robustness of the proposed method to the choice of the dataset. Consequently, I recommend rejecting the paper and propose authors to resubmit after adding more datasets as part of their evaluation.